# Evaluating altimetry-derived surface currents on the South Greenland Shelf with surface drifters

Arthur Coquereau[1,2] and Nicholas P. Foukal[1]

[1]Woods Hole Oceanographic Institution, Woods Hole, MA 02543, USA
[2]Laboratoire d'Océanographie Physique et Spatiale, Univ. Brest CNRS IRD Ifremer, Brest, France

**Correspondence:** Arthur Coquereau (arthur.coquereau@univ-brest.fr)

**Abstract.** The pathways and fate of fresh water in the East Greenland Coastal Current (EGCC) are crucial to the climate system. The EGCC transports large amounts of fresh water in close proximity to sites of deep open ocean convection in the Labrador and Irminger Seas. Many studies have attempted to analyze this system from models and various observational platforms, but the modeling results largely disagree with one another, and observations are limited due to the harsh conditions typical of the region. Altimetry-derived surface currents, constructed from remote-sensing observations and applying geostrophic equations, provide a continuous observational data set from 1993. However, these products have historically encountered difficulties in coastal regions, and thus their validity must be checked. In this work, we use a comprehensive methodology to compare these Eulerian data to a Lagragian data set of 34 surface drifter trajectories and demonstrate that the altimetry-derived surface currents are surprinsigly capable of recovering the spatial structure of the flow field on the South Greenland Shelf and can mimic the Lagrangian nature of the flow as observed from surface drifters.

## 1 Introduction

Along the East Greenland Shelf, meltwater from both the Arctic pack ice and the Greenland Ice Sheet flow southward toward the southern tip of Greenland (Foukal et al., 2020). This southward flow is concentrated in two current cores that are each supported by bathymetric gradients: the East Greenland Current (EGC) at the shelfbreak, and the East Greenland Coastal Current (EGCC) at the coastline (Håvik et al., 2017). Around Greenland, the deep shelves (150-500 m), steep gradient at the coastline, and strong along-shelf barrier winds sustain this two-core system. If the buoyant meltwater water masses in the EGCC are mixed offshore, they may stratify regions of deep water formation and water mass transformation, inhibit air-sea heat exchange, and slow the Atlantic Meridional Overturning Circulation (AMOC; Rahmstorf et al., 2015; Hansen et al., 2016; Glikson, 2019).

As these currents approach Cape Farewell at the southern tip of Greenland, strong northerly barrier winds on the Southeast Greenland Shelf push the fresh, surface waters toward the coastline and into the EGCC (Duyck and De Jong, 2021). At Cape Farewell, the downwelling-favorable barrier winds relax as they meet the northerly, upwelling-favorable winds from the Southwest Greenland Shelf (Pacini and Pickart, 2023). The confluence of these wind regimes at Cape Farewell leads to the mean winds crossing the shelf and potentially allowing large amounts of fresh water to leave the shelf here.

Many modeling studies have analyzed this potential for shelf-basin exchange around Cape Farewell (Fichefet et al., 2003; Marsh et al., 2010; Dukhovskoy et al., 2016; Böning et al., 2016; Gillard et al., 2016; Luo et al., 2016; Schulze Chretien and Frajka-Williams, 2018; Oliver et al., 2018; Dukhovskoy et al., 2019; Pennelly et al., 2019; Castelao et al., 2019; Garcia-Quintana et al., 2019; Tagklis et al., 2020; Dukhovskoy et al., 2021; Gou et al., 2021, 2022; Duyck et al., 2022; Morrison et al., 2023). There are many conflicting results between these various publications, notably some detect a very clear signal

of Greenland meltwater influencing the large-scale circulation (Garcia-Quintana et al., 2019), while others document a more muted oceanic response to the freshwater forcing (Böning et al., 2016). Dukhovskoy et al. (2019) note the importance of the vertical mixing rate in the effect of this freshwater, and the rapid dilution of the meltwater signature if mixed below the upper 100 m. In contrast, Tagklis et al. (2020) and Pennelly et al. (2019) find a clear relationship with freshwater input and shoaling of the mixed layer depths in the Labrador Sea. Luo et al. (2016) and Gillard et al. (2016) both note the strong sensitivity to where

the meltwater is introduced from Greenland to the shelf, with meltwater introduced on the East Greenland Shelf more likely to leave the shelf in the Labrador Sea than meltwater introduced on the West Greenland Shelf. Schulze Chretien and Frajka-Williams (2018) and Duyck et al. (2022) report an Ekman, wind-driven mechanism to transport freshwater off the Greenland Shelf, while Tagklis et al. (2020) note the importance of boundary current instabilities in driving the shelf-basin exchange.

       Reconciling these various model results requires an observational data set of ocean currents that covers similar geographic

extents as the models. Satellite altimetry is an example of such a dataset. Though each individual pass of a nadir-altimeter satellite measures just a single line of sea-surface height (SSH) directly below the satellite's path, the combination of multiple satellites that cross a region can yield significant data coverage, even over smaller regions such as the southern tip of Greenland. These individual passes are then gridded onto a ¼° global grid (Rio et al., 2011) to yield a spatially contiguous, and temporally continuous data set of SSH. The pressure gradients resulting from structure in the SSH field are then used to calculate surface

geostrophic velocities (Rio et al., 2011). In addition, Ekman velocities derived from atmospheric reanalyses are combined with the geostrophic velocities to yield a surface velocity that more accurately accounts for the large-scale wind driven flow (Rio et al., 2014). It is often assumed that the gridded SSH data do not perform well on the continental shelves due to the smaller scale features in the coastal circulation, as well as errors in the altimetry associated with sampling close to coastlines (Pujol et al., 2023). The Greenland shelf is unique in this regard though because it is deep (150-500 m), the bathymetric gradient from

the coast to the shelf is steep, and the shelf circulation is vigorous (velocities on the order of 0.4-1 m/s). Thus determining whether altimetry-derived surface currents (ADSC) in the region are valid is of interest.

       In August 2021, we deployed 38 surface drifters on the Southeast Greenland Shelf to determine the pathways of the shelf circulation around Cape Farewell. This observational campaign provides an unprecedented view into the surface circulation of the region, though it is limited to about two weeks during which the drifters passed through the area. Here, we use these

drifters to evaluate the performance of the ADSC and assess whether the gridded altimetry data product is suitable for longer-term studies. The paper is structured as follows: in section two, we describe the data sets and a novel three-step methodology to robustly compare Lagrangian and gridded Eulerian data. In section three, we present the results of this methodology. And in the fourth section we discuss the results in light of other published literature and implications for future work.

# 2 Data and Methods

## 2.1 Description of surface drifters

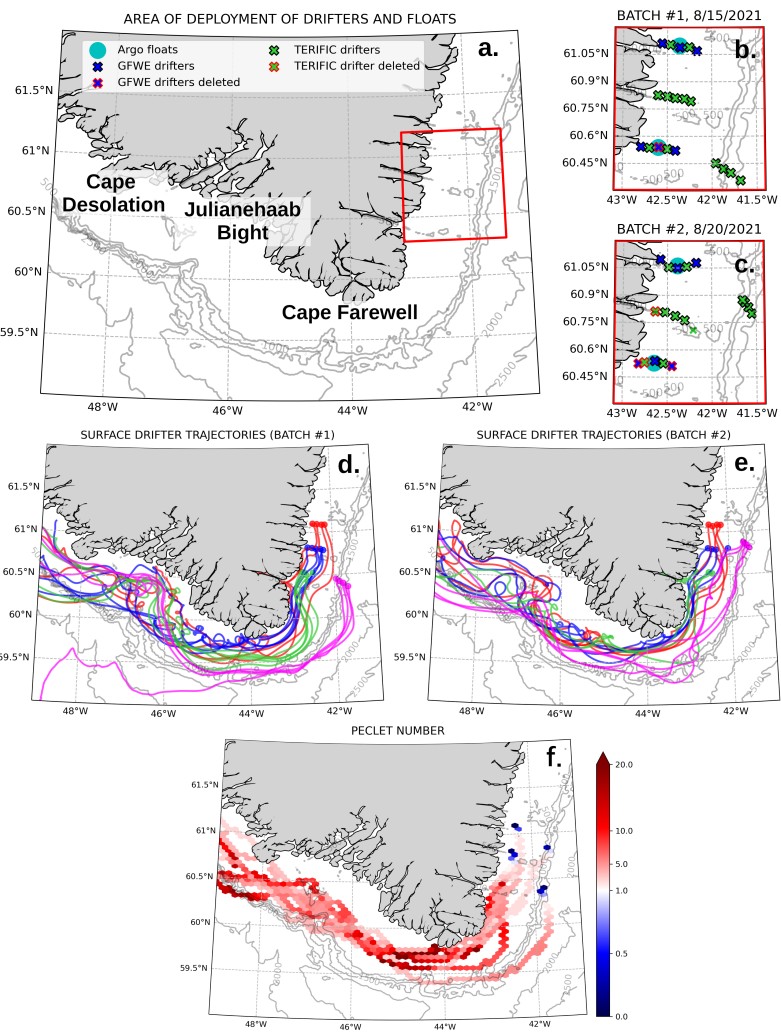

**Figure 1.** (a) The study region, with Cape Farewell at 44°W, Julianehaab Bight at 46°W and Cape Desolation at 48°W. Contour lines represent isobaths. Deployment location of drifters and floats respectively for first (b , August 15th) and second (c , August 20th) batch. Drifter trajectories respectively for first (d) and second (e) batch. Colors represent lines of drifters. (f) Péclet number derived from the lines of drifters, representing the ratio advection over dispersion. The trajectories of the centroids of each line are represented, overlaping data are averaged. The calculation of Péclet number is detailed in section 2.1.

In August 2021, 38 SVP drifters were released on the southeastern shelf of Greenland (Fig. 1 a) from R/V *Neil Armstrong*. These surface drifters are drogued at 15 m with a drogue-to-buoy drag ratio of 40:1 to ensure drifters follow the 15 m current

and are not directly pushed by the wind. Two distinct programs provided these drifters: the National Science Foundation (US) funded Greenland Fresh Water Experiment (GFWE) with 12 drifters and the European Research Council (EU) funded

Targeted Experiment to Reconcile Increased Freshwater with Increased Convection (TERIFIC) with 26 drifters. The drifters were separated evenly into two batches of 19 drifters that were deployed five days apart (August 15th and 20th). Each batch consisted of one line of four drifters deployed at the shelfbreak, and three zonal lines of five drifters deployed across the inner shelf. The three lines were separated meridionally by 25-30 km between one another, and the drifters along each line were separated by about 5 km (Fig. 1 b, c).

A number of parameters were then checked. The continuous presence of the drogue was verified using two methods: (1) the buoy parameters, (2) the coefficient of determination $R^2$ between complex drifter velocity and complex wind velocity (from ERA5, Hersbach et al., 2020) or similarly the least-square complex linear regression of drifter velocity from wind velocity (Kundu, 1976; Poulain et al., 2009). Both methods confirmed that all drifters maintained their drogues throughout the period of analysis described in this paper. We also tested whether any of the data points qualified as outliers according to the Elipot et al. (2016) definitions, but we did not identify any. Four of the drifters were trapped in fjords for prolonged periods, thus we did not consider them further. These steps left us with 34 trajectories to analyze. We then filtered out high frequency variability using a 24 hr cut-off low-pass filter. More details on drifter data processing are available in Section 1 of the supplementary material.

To assess the relative importance of advection compared to dispersion, we calculated the Péclet number (Fig. 1 f) from the drifter dataset. This is calculated by considering each line of drifters as a group. For each line and time step, the dispersion of the drifters is computed as the root mean square distance of drifters from the center of gravity of the line. The advection is estimated from the distance traveled by the center of mass of the line since the previous time step. The advection is then divided by the dispersion to obtain the Péclet number. Values greater than 1 indicate the advective component is dominant.

## 2.2   Description of satellite altimetry data sets

Two data sets of ADSC have been evaluated in this study. The first is the Copernicus Marine Environment Monitoring Service (CMEMS) near-real time product denominated "Global Ocean Gridded L4 Sea Surface Heights And Derived Variables Nrt". The processing used is the DUACS multi-mission altimeter data processing system DT-2021 (Faugère et al., 2022) provided by CNES/CLS with a methodology detailed in Pujol et al. (2016). The altimetry data is merged from all available altimetry missions (Fig. S1 from supplementay material) and interpolated on a 1/4° grid with a daily resolution. The geostrophic currents provided in this data set benefit from the CNES-CLS18 Mean Dynamic Topography (MDT) product (Mulet et al., 2021). This MDT gathers altimetry, gravity and drifter data. It shows better results in all regions around the globe compared to the previous product CNES-CLS15 especially in coastal areas. The geostrophic currents are computed using a nine-point stencil width methodology (Arbic et al., 2012). In the following work, we refer to this data set as "Geostrophy".

The second ADSC data set is the "Global Total Surface and 15m Current (COPERNICUS-GLOBCURRENT) from Altimetric Geostrophic Current and Modeled Ekman Current Processing". The Ekman velocities at 15 m depth are computed from ERA5 (Hersbach et al., 2020) 3h wind data following the methodology developed in Rio and Hernandez (2003), Rio et al.

(2011), Rio (2012) and Rio et al. (2014). An empirical Ekman spiral-like model is estimated, based on 2 parameters determined from a least-squares regression from SVP drifters' data, Argo floats data and wind stress measurements from ECMWF ERA5. We refer to this data set as "Geostrophy+Ekman" because it is the addition of Ekman contribution to the previous geostrophic current data set. The final product provides a 6-hour frequency data set with zonal and meridional components of surface velocity. For fair comparison with the daily Geostrophy product, here we primarily evaluate daily velocities but have also looked at the 6-hour frequency product with higher frequency wind data to see the impact of higher temporal resolution. The 15 m depth velocities are used for consistency with 15 m depth drogued drifter.

In addition to the two ADSC data sets, we also compare the drifter velocities to the "Arctic Ocean Physics Analysis and Forecast" product from the operational TOPAZ4 Arctic Ocean system (Sakov et al., 2012) with the same 1/4° resolution. This is a completely different data set in that it is a model that assimilates all possible data streams (e.g. Argo, satellites) rather than only relying on satellite altimetry and wind products. The goal of including TOPAZ in the first step of this methodology is to provide context to the results from the ADSC data sets. This data assimilation product is based on the HYCOM model and a 100-member Ensemble Kalman Filter (EnKF) assimilation scheme.

## 2.3 Motivation for a framework to compare Lagrangian and gridded Eulerian velocity fields

The goal of this paper is to directly compare the velocity fields derived from the surface drifters with those derived from altimetry. But direct comparisons between Eulerian and Lagrangian data is difficult because the reference frames provide fundamentally different information about the flow field. Eulerian data inform how the flow field evolves through time, while Lagrangian data provide information on the origin, pathways and fate of fluid particles. The spatial scales also typically vary between these two reference frames. While Eulerian data can come from point measurements or data mapped onto a regular grid that typically cover larger spatial scales, Lagrangian data only come from point measurements, typically on smaller scales. Comparing point measurements from Eulerian and Lagrangian reference frames is not useful in most applications – these two data streams are only directly comparable when they physically intersect, and when they do, it is straightforward to compare them. Thus, we do not discuss this comparison any further. In contrast, the need to compare gridded Eulerian data with Lagrangian data arises quite often in oceanography. The proliferation of gridded satellite and reanalysis products, as well as numerical model output has produced a large number of gridded Eulerian data sets (e.g., Haine et al., 2021). These data sets are often compared to surface drifters from the Global Drifter Program (GDP; Lumpkin and Johnson, 2013) and profiling floats from the Argo network (Johnson et al., 2022) to yield a velocity field that utilizes the accuracy of the in situ Lagrangian data with the spatial perspective of the gridded Eulerian data.

Though the need for a robust framework for comparing gridded Eulerian and Lagrangian velocities exists, there is not yet a well-accepted, published methodology for these comparisons. Previous work on this topic typically converts one of the data sets into the other reference frame, then applies standard statistical methods such as correlation or variance metrics (Liu and Weisberg, 2011; Liu et al., 2014; Rio et al., 2011; Pujol et al., 2016; Rio and Santoleri, 2018; Mulet et al., 2021). This can be done by either gridding Lagrangian velocities onto a Eulerian grid or simulating Lagrangian trajectories through a Eulerian velocity field. While both methods yield data sets that can be directly compared, the process of transforming the data between

reference frames inevitably degrades the converted data by interpolating or extrapolating when/where data is not available. If this comparison is only done in one direction, this process likely biases the results toward one reference frame. Thus, to robustly compare Eulerian and Lagrangian data, this conversion must proceed in both directions.

## 2.4 Comparison framework

In this work, we propose a framework from which Eulerian and Lagrangian velocity fields can be directly compared. The framework directly addresses three questions that leverage the relative strengths of each perspective: (step #1) How well do the gridded Eulerian velocities resolve the velocities directly observed by the Lagrangian platform? (step #2) How well do the Lagrangian data recover the spatial structure of the gridded Eulerian data? (step #3) How well do the Eulerian data capture the origins, pathways, fate, and connectivity of water masses? To answer these questions, we use three analysis steps: (step

#1) directly compare the velocity fields as measured at collocated and contemporaneous points, (step #2) map the Lagrangian velocities onto a Eulerian grid, and (step #3) simulate Lagrangian trajectories in the Eulerian flow fields. The first step directly compares the two data sets without any transformation between reference frames, while the second two steps convert one data set into the other's reference frame. All three steps utilize the strengths of the data sets: the accuracy of the Lagrangian data (step #1), the spatial perspective of the Eulerian data (step #2), and the origins, pathways, fate, and connectivity of the

Lagrangian data (step #3).

### 2.4.1   Point-wise comparison between gridded Eulerian and Lagrangian velocity fields

    The point-wise method consists of extracting the velocities of the gridded Eulerian field along the Lagrangian trajectories (Fig. 2) and comparing them to the Lagrangian velocities. To select the collocated and contemporaneous node on the ADSC grid that matches a specific spatio-temporal drifter data, we extracted the nearest grid point from the drifter location corre-

sponding at the same time step. It is also possible to use other interpolation method such as linear interpolation or optimal interpolation, though we found that using the nearest grid cell retained more of the variance in the ADSC, and was thus preferable to interpolation schemes. Here, the point-wise method has been performed on drifter daily velocities to extract the collocated and contemporaneous velocities from the ADSC. Four components of the velocities have been evaluated: the zonal (u), the meridional (v), the along-shelf, and the across-shelf velocities. These two latter velocity components, which are simply

projections of the velocity vectors according to a geographic feature (here, the shelfbreak) tell us how the velocity field moves independently from the shape of the bathymetric gradients at the shelfbreak. Practically, this involves first defining a smoothed shelfbreak by lowpass filtering the 600 m isobath to remove the small scale features such as submarine canyons and troughs. The cross-shelf vector ($z_{cross}$, in complex form) is obtained by connecting a straight line from the surface drifter to the closest point of the shelfbreak. The along shelf angle is defined to be perpendicular to the across shelf angle. Fig. S2 provides a visual

representation of this coordinate transformation. The velocity vector ($z_{velocity}$) at the position of the surface drifter is computed as $z_{vel} = u + iv$. The angle $\theta$ between the cross-shelf vector and the drifter velocity vector is computed as:

$$\theta = \arg\left(\frac{z_{cross}}{z_{velocity}}\right)$$

$$v_{along} = \sin\theta.|z_{velocity}|$$

$$v_{across} = \cos\theta.|z_{velocity}| \tag{1}$$

Once the velocities have been rotated, we extract a time series of ADSC velocities along each drifter trajectory. For each drifter, we compare, the standard deviation (normalized by drifter velocity standard deviation, $\hat{\sigma}_f$), the correlation coefficient $r$ relative to the drifters' daily velocities, the root-mean-square error (RMSE) and the percent variance explained between the two time series (equ. 2).

$$\text{Variance of X explained by Y } (\%) = 100 \times \left[1 - \left(\frac{\sigma^2(X-Y)}{\sigma^2(X)}\right)\right] \tag{2}$$

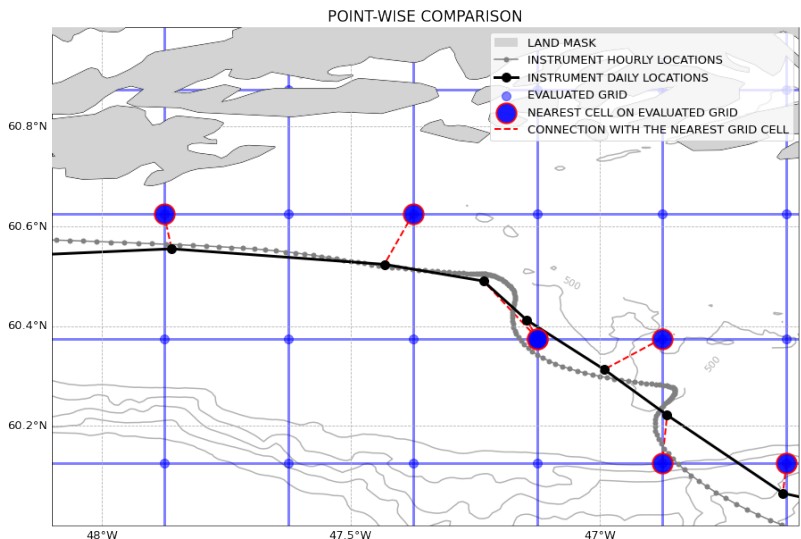

**Figure 2.** Representation of the point-wise comparison. The hourly Lagrangian trajectory (gray) is averaged to daily resolution (black) to match the resolution of the altimetry-derived surface currents (ADSC) field. The Lagrangian velocity at each daily location is calculated as the distance between the location 12 hr before the daily location and the location 12 hr after, divided by 24 hr. The nearest grid cells of the Eulerian product (blue dots with red edges) are determined along the Lagrangian trajectory and the velocity at those points is compared to the Lagrangian velocity. The grey areas represent the coastline, and the contours represent the bathymetry.

Taylor diagrams and skill scores $S$ (equ. 3) (Taylor, 2001) are used to concatenate and summarize the standard deviation, correlation coefficient, and RMSE. We compare these skill scores to the TOPAZ4 reanalysis product, which serves as a baseline from which to compare the ADSC values.

$$S = \frac{4(1+r)}{(\hat{\sigma}_f + 1/\hat{\sigma}_f)^2 . (1+r_0)}$$

(3)

with $r_0$ the maximum expected correlation coefficient (here taken equal to one).

### 2.4.2 Eulerian gridding of Lagrangian velocities

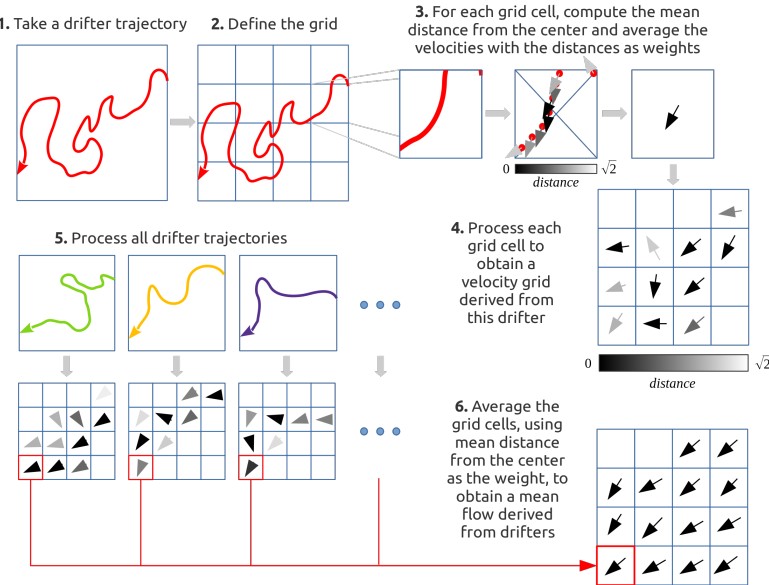

**Figure 3.** Schematic of the averaging of velocities. This method is intended for Lagrangian instruments, but steps 4-6 can be applied to the Eulerian product using the velocity at the nearest grid cell and corresponding in time to the Lagrangian location.

The Eulerian gridding consists of mapping the Lagrangian data into a Eulerian gridded field. The gridded field is obtained by averaging the Lagrangian velocities on the same $1/4°$ grid as the ADSC.

Rather than the classical averaging of all Lagrangian devices available together in the same grid, we use an alternative method that corrects for the number of data points reported in a given area by slowly and quickly moving instruments. Otherwise, slow instruments have more position points in each bin and thus exert a stronger influence on the average.

To perform the mapping (Fig. 3), we proceed sequentially through each individual drifter and produce a unique gridded map based on data from a single drifter. For each Lagrangian trajectory s, all the cells are scanned to check if the drifter has passed inside. If the drifter has not passed through the cell, it is left empty. Otherwise, all velocities within the cell are averaged. This average is weighted to attribute more weight to the observations close to the center using a Gaussian distribution.

The grid $U$ obtained for each Lagrangian instrument therefore contains empty cells (where it has not passed) and cells containing the average velocity measured inside (equ. 4). A similar grid $D$ is filled with the average distances from the center

of the cells calculated using the same method (equ. 5). This processing is repeated for each of the 34 drifters and the velocities across the drifters are averaged together according to their weighted distances from the center of the cell (equ. 6).

Mathematically, this process can be explained by taking $u(s, x)$, the velocity of the instrument $s$ at location $x = (x, y)$. The gridded velocity field $U(s, c)$ of the Lagrangian trajectory $s$ at the grid cell $c$ located at $x_c = (x_c, y_c)$ is computed as:

$$U(s, c) = \frac{\sum^x u(s, x).q(s, x)}{\sum^x q(s, x)} \tag{4}$$

And the grid of mean distances $D$ is computed as:

$$D(s, c) = \frac{\sum^x d(s, x).q(s, x)}{\sum^x q(s, x)} \tag{5}$$

with $x = (x, y)$ and only if $x \in \left[x_c - \frac{\Delta x}{2}; x_c + \frac{\Delta x}{2}\right] \bigcap y \in \left[y_c - \frac{\Delta y}{2}; y_c + \frac{\Delta y}{2}\right]$

$\sum^x$ means sum all locations verifying the previous condition (selecting only locations in the cell $c$), $q(s, x)$ is the weight of the location $x$ of instrument $s$ depending on the distance from the center with, as an example, a gaussian weighting like:

$$195 \quad q(s, x) = exp\left(-\frac{d^2}{0.5\sqrt{(\Delta x)^2 + (\Delta y)^2}}\right) \tag{6}$$

This processing is repeated for each of the drifters and the 34 grids are then averaged. For each cell, the average of the 34 grids in that cell is calculated (equ. 7) and weighted by the averaged distances to the center (equ. 8). The final Eulerian velocity grid $\mathcal{U}$ combines the 34 grids and thus the trajectories of all the drifters.

$$\mathcal{U}(c) = \frac{\sum_{s=1}^{n} U(s, c).Q(s, c)}{\sum_{s=1}^{n} Q(s, c)} \tag{7}$$

$$200 \quad Q(s, c) = exp\left(-\frac{D(s, c)^2}{0.5\sqrt{(\Delta x)^2 + (\Delta y)^2}}\right) \tag{8}$$

The velocity field is not entirely covered by Lagrangian instruments at all times. Our drifters were initially located on the Southeast Greenland Shelf, and moved progressively around Cape Farewell to the west. So the drifters did not sample the Southwest Greenland Shelf initially and the Southeast Greenland Shelf near the end of the two-week period. To properly account for this heterogeneous sampling of the drifters, we constructed a comparison data set by sub-sampling the ADSC

fields in a similar fashion as the drifters sampled the real ocean. This procedure accounted for the temporal biasing that often occurs when comparing in situ observations with other more homogenously sampled data sets. To assess whether our results were sensitive to this procedure, we compared the results with this sub-sampled ADSC product with the mean of the entire

ADSC fields over the same time period and they were nearly identical. The gridded Lagrangian and Eulerian products are then compared to evaluate the differences relative to spatial patterns of velocity and directions and magnitudes of currents.

As pointed out by LaCasce (2008) the choice of bin size in this type of analysis is not trivial. Bins that are too large will smooth out the field and bins that are too small will make the mean very sensitive to eddies and other fine-scale structures. In most of the cases, a convenient choice to facilitate the comparison is to use an identical grid as the original Eulerian gridded products. However, to evaluate the influence of gridding and the consistency of the 1/4° drifter gridding with the original drifter data we compared the results at 1/4° resolution to 1/12°. This higher resolution was chosen to test the sensitivity of the results to the resolution of the grid and ensure that an average of 4 drifters passed through each cell. A higher resolution would imply fewer drifters on average. We also tested another gridding methodology based on a k-mean clustering as proposed by Koszalka and LaCasce (2010) that has the advantage of directly use the density of data to determine the location of grid nodes and does not require arbitrary choices regarding the grid resolution and bounds. However, due to the heterogeneous distribution of drifters on the region, the clustering method led to spurious results.

### 2.4.3 Observed and synthetic trajectories

The third part of the comparison framework consists of computing synthetic Lagrangian trajectories from the ADSC Eulerian field (Fig. 4). This corresponds to converting the Eulerian data into the Lagrangian frame of reference. The main idea is to compute a synthetic trajectory by advecting a Lagrangian particle through the Eulerian velocity field (Liu and Weisberg, 2011). We use the Parcelsv2.0 (Delandmeter and van Sebille, 2019) particle tracking software to simulate the trajectories. The resulting synthetic trajectories must then be compared to the observed trajectories of drifters and evaluated using a metric. Various methods could be used such as the distance between the ending points of trajectories, the main direction of trajectories, or a cloud of points, etc. We have found the methodology proposed by Liu and Weisberg (2011) particularly instructive: at each location of the Lagrangian data, a new synthetic particle is released and advected through the velocity field for three days. Then, each three-day trajectory is compared to the actual trajectory over those three days, and a skill score is assessed for each particle launch. This skill score (equ. 9, 10) combines the cumulative distance traveled by observed particles (Lagrangian devices) $dl_i$ and the cumulative separation distance between synthetic and observed particles $d_i$. It is important to note that this skill score is quite rigorous; to produce a perfect skill score (equal to one), the simulated particle must not only end up at the same location after three days, but also follow the exact pathway to the end point.

$$c = \frac{\sum_{i=1}^{3} d_i}{\sum_{i=1}^{3} (\sum_{j=1}^{i} dl_j)} \tag{9}$$

$$s = \begin{cases} 1 - c, & \text{if } c \leq 1 \\ 0, & \text{otherwise} \end{cases} \tag{10}$$

A disadvantage of this metric is its sensitivity to the duration of the advection of the particles. As proposed by Liu and Weisberg (2011), we tested the sensitivity of our results to the duration of the particles' advection by repeating the experiment

with various durations spanning from 2 to 14 days. We selected three days because it identified differences in trajectories over the synoptic wind forcing time scale in the region. Another potential issue with this skill score derived in Liu and Weisberg (2011) is that it sets negative skill score values to zero, and alias regions of poor skill toward better scores. One example of an alternative skill score is introduced by Révelard et al. (2021) as nearly identical to that in Liu and Weisberg (2011) but retains the negative skill scores. We compare both metrics and do not identify large differences.

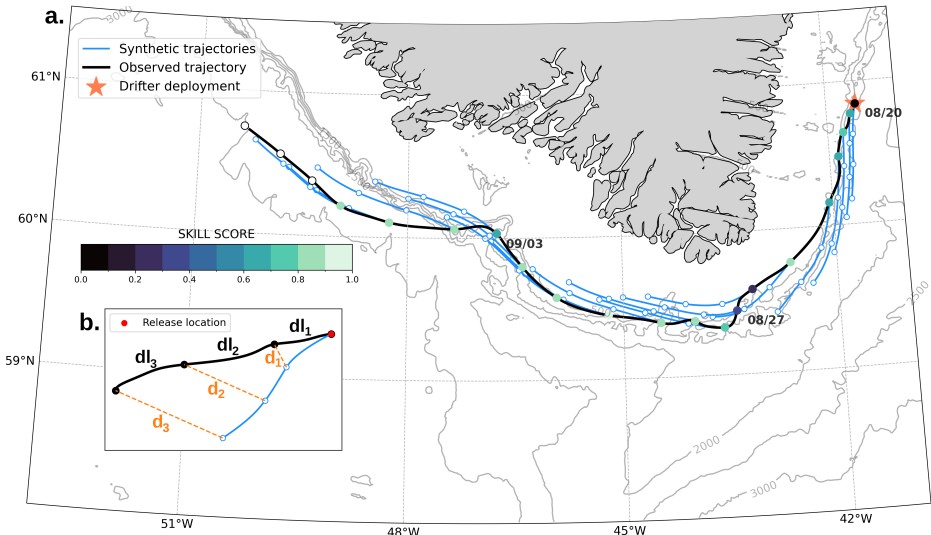

**Figure 4.** Illustration of the method involving a comparison between observed and synthetic particles using the approach of Liu and Weisberg (2011). (a) Skill scores (colored circles) calculated along a Lagrangian instrument trajectory (black line). Blue lines represent synthetic particle trajectories run over three days. Dot color represents skill score at the daily observed location where the synthetic particle has been released. The orange star represents the location where the drifter was released. Grey contours are isobaths. (b) A schematic of the various variables used to calculate the skill score.

## 3 Results

### 3.1 Applying the point-wise comparison to surface drifters and ADSC around Cape Farewell

We present here the results of the point-wise comparison between the surface drifters and ADSC around Cape Farewell. Figure 5 shows the Taylor diagrams of the evaluated products (ADSC and TOPAZ4). Differences between high scores are visually exaggerated with non-linear axes in order to detail differences between the products. Meridional velocities (v) are well resolved by the altimetry-derived products (Fig. 5 b) with an average skill score of 0.87 computed from the average correlation coefficients of 0.75 (Geo.) and 0.79 (Geo.+Ekman) and the average normalized standard deviations of 0.84 (Geo.+Ekman) and 0.92 (Geo.). The zonal component (u) has lower scores overall (Fig. 5 a) with average skill scores of 0.63 (Geo.) and 0.69 (Geo.+Ekman) computed from the average correlation coefficients of 0.62 (Geo.) and 0.64 (Geo. + Ekman) and the average normalized standard deviations of 0.60 (Geo.) and 0.65 (Geo.+Ekman). This difference in scores between the u and

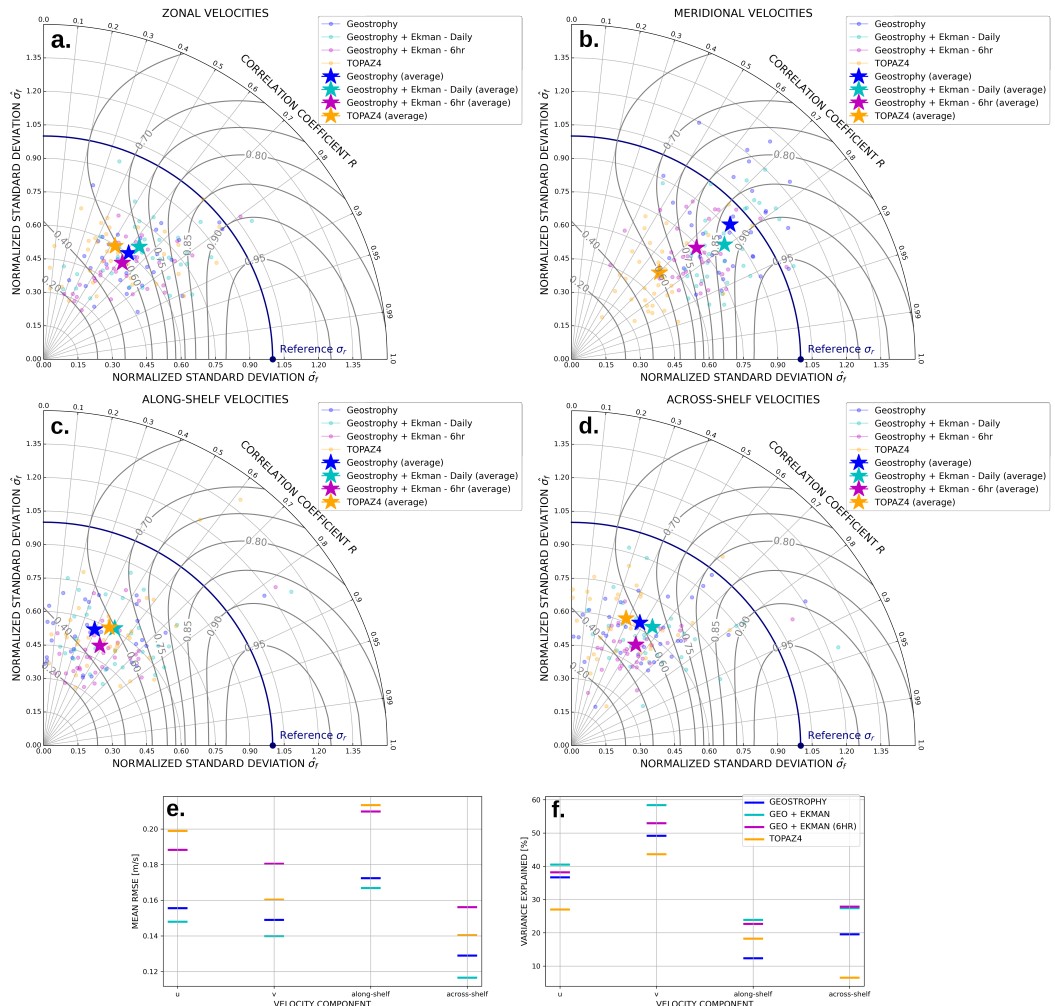

**Figure 5.** Taylor diagrams of the point-wise comparison between the surface drifter velocities and either ADSC or TOPAZ4 for the (a) zonal, (b) meridional, (c) along-shelf and (d) across-shelf components. The x and y axes depict the normalized standard deviation (ratio of ADSC's standard deviation to surface drifter's standard deviation). Polar coordinates represent the correlation coefficient between product and drifter timeseries. Contours represent Taylor Skill Scores. Each trajectory for each product is represented with a point. The stars represent the average across all 34 trajectories of normalized standard deviations and correlation coefficients for each product. Note the non-linearity of scales used for correlation and skill score contours in the figure. The Taylor diagrams are modified versions of an initial Python code by Copin (2012). (e) Root mean square error (RMSE) for each velocity component. (f) Percentage of variance explained for each velocity component.

v components can be explained by the shape of the coast that makes meridional velocities easier to reproduce. As the drifters follow the shelf around Cape Farewell the meridional velocities shift from high negative velocities (southward) to positive velocities (northward). If the gridded Eulerian product simulates this general behavior, i.e. the current follows the bathymetry,

then the skill scores of the meridional velocities will be higher. Another potential explanation of this difference is that sea surface height gridded at 1/4° used to compute geostrophic currents, does not have the same spatial resolution in latitude and in longitude. A degree of latitude corresponds always to the same distance, $\approx 111$ km, but a degree of longitude at 60°N corresponds to half the latitudinal resolution, or 55 km. In a 1/4° product, the grid cells are 27 km in latitude, and 13 km in longitude. The Rossby Radius in this region is on the order of 10 km, thus the 1/4° product approaches the Rossby Radius in longitudinal span. Meridional geostrophic velocities, computed as $v_g = \frac{g}{f} \frac{\partial \eta}{\partial x}$, benefit from the better longitudinal resolution. The difference between altimetry-derived velocity and reanalysis product is very important for meridional velocities, though it should be noted that the velocities in TOPAZ4 do not experience such improvement from u to v (average skill score for zonal component is 0.59 and for meridional, 0.60).

The along and across-shelf velocities present lower average Taylor skill scores (Fig 5 c and d), respectively 0.51 (Geo.), 0.60 (Geo.+Ekman) and 0.60 (Geo.), 0.64 (Geo.+Ekman). This decrease from u and v to along-shelf and across-shelf velocities is due to the fact that coordinate system follows the bathymetry and thus only considers variability distinct from the bathymetric contours. The across-shelf Taylor diagram shows that Geostrophy+Ekman performs better than other products especially compared to Geostrophy only.

The RMSE calculated for each velocity component shows very good results for Geostrophy compared to TOPAZ4 (Fig. 5 e). The RMSE was much lower for the across-shelf component especially compared to the along-shelf one – the across-shelf velocities are often difficult to model with geostrophy alone and thus the expectation is that the error should be higher. However, this lower RMSE could be explained by the smaller magnitude of the across-shelf velocities, which imply a lower signal-to-error ratio. This hypothesis is reinforced by the percent variance explained (Fig. 5 f) that normalizes these errors by the total amount of variance in each direction. When the amount of total variance is accounted for, the along-shelf and across-shelf metrics are similar, and the difference between the u, v fields and the along and across-shelf velocities is accentuated. In addition, the effect of the Ekman component is more apparent in the percent variance explained plot. In the along-shelf direction, the addition of the Ekman component nearly doubles the percent variance explained. Similarly, in the across-shelf direction, the Ekman component makes an important contribution but surprisingly, the change in percent variance explained when the Ekman component is added is larger in the along-shelf direction than the across-shelf direction. Given that the flow along bathymetric contours (along-shelf direction) is largely thought to be geostrophic, while deviations to it (the across-shelf direction) would be ageostrophic, it is surprising that the Ekman component is more important in the along-shelf velocities than the across-shelf velocities. The Ekman component can influence the along-shelf currents by winds that are misaligned with the currents. This misalignment occurs frequently in the vicinity of Cape Farewell as the dominant wind patterns are no longer constrained by the topography of Greenland and the winds cross the shelf. Thus there is good reason for why the Ekman contribution is large in the along-shelf direction.

The Geostrophy+Ekman data set are also available at 6-hour frequency due to the higher frequency wind products, so we also investigated the difference between the 6-hourly data and the 6-hourly drifter data. Recall that we chose the daily Geo.+Ekman data for more direct comparison to the Geostrophy only fields, which are only available at daily resolution. The 6-hourly Eulerian data compared to 6-hourly drifter data have comparable correlation coefficients but weaker normalized

standard deviations than the other altimetry-derived products. This data set has also a larger RMSE (Fig. 5 e) but the percent variance explained is close to the daily data. Thus the higher temporal resolution does not directly lead to better results because the 6 hourly drifter data contains more variability. However, the higher temporal resolution is likely more representative of the total velocity fields, and thus preferable for further studies.

Some drifters tracks scored very poorly in the Taylor diagrams, reflecting the fact that it is difficult for the ADSC to reproduce exactly the same velocities as those experienced by drifters. Indeed, some of them may for example encounter very small scale features near the coast or due to bathymetric changes that may lead to different velocities than the main current. However, we also observed some trajectories with very high scores and a good general representation of the velocities summarized through the mean scores.

We tested the sensitivity of our results to the interpolation methods. When comparing the 'nearest neighbor' method with linear interpolation, the correlation improved on average by 0.02, and the percent variance explained increased by 2%, but the normalized standard deviation and the Taylor Skill Score both decreased by 0.03 in average. Thus interpolating reduced the variance in the gridded data while improving the correlations. The gridded altimetry is already a smoothed product from the along-track altimetry. We felt that retaining the natural variance is important. Given the similarity between the results, our
conclusions from this first step are robust to this choice.

### 3.2    Eulerian gridding of surface drifter velocities and ADSC

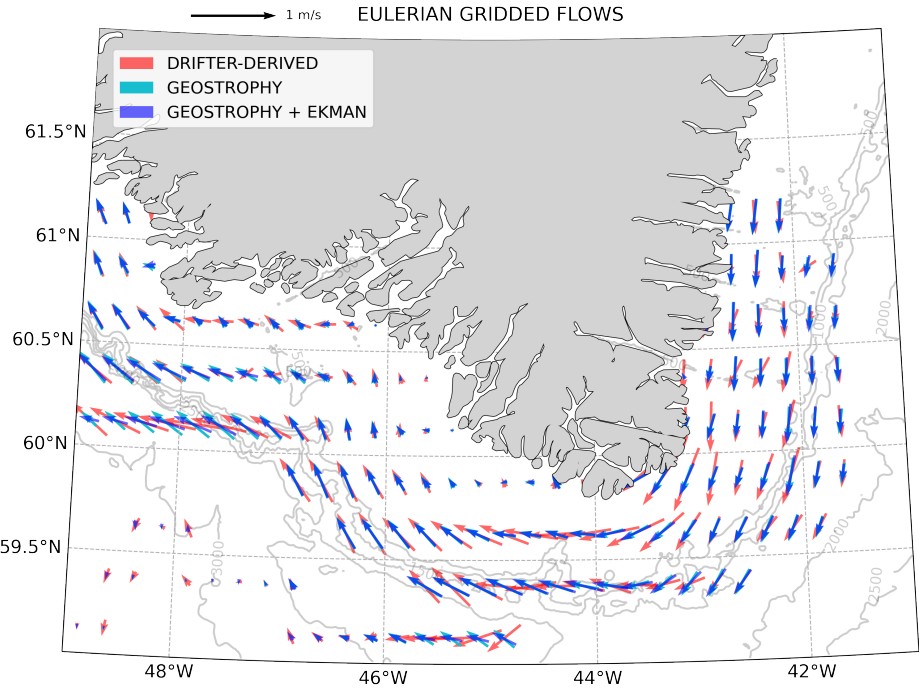

**Figure 6.** Gridded surface currents derived from drifter data (red), Geostrophy product (cyan) and Geostrophy+Ekman product (blue).

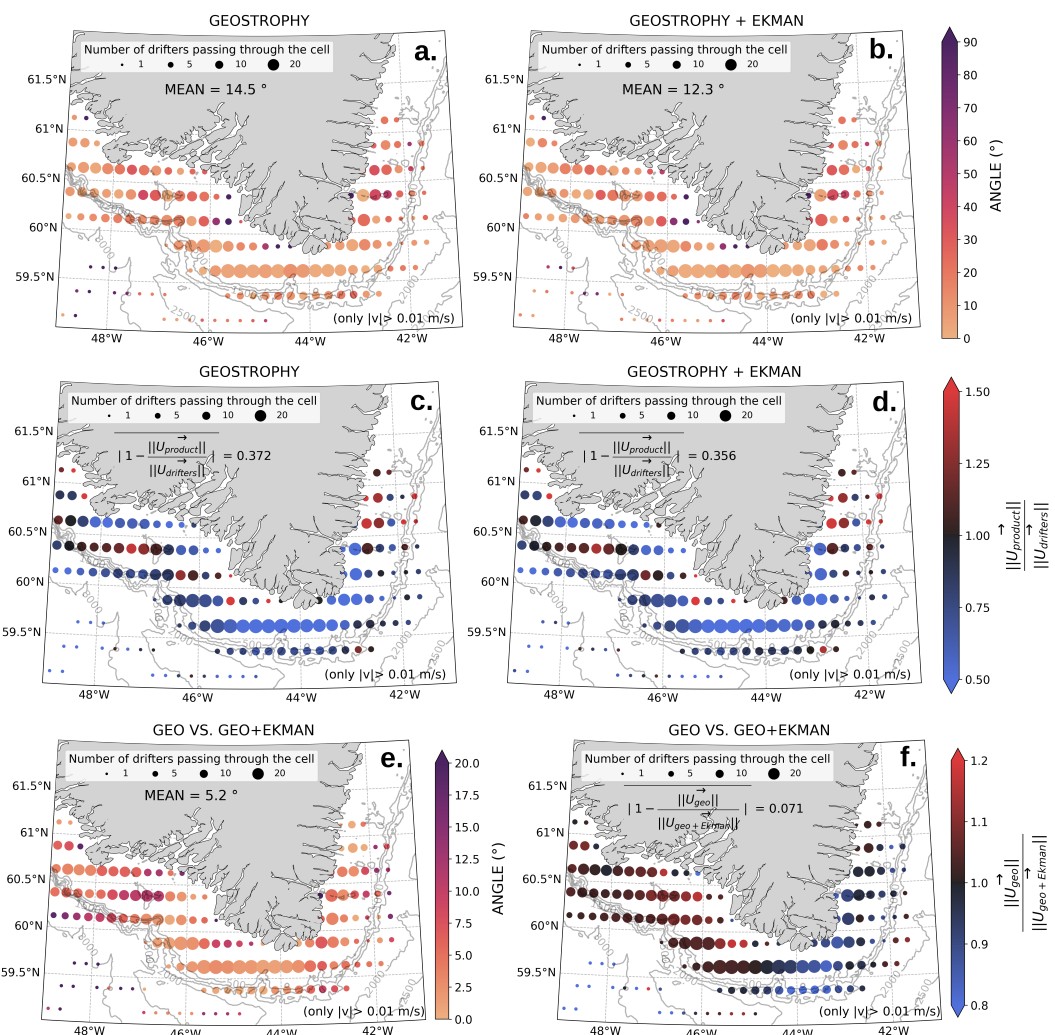

**Figure 7.** Differences in angle (a, b) and magnitude (c, d) between surface drifter velocities and Geostrophy (a, c) and Geostrophy+Ekman (b, d) velocities. In panels a and b, color represents the angle and size of dots represents the number of different drifters passing through a cell. Only velocities greater than 1 cm/s are considered to ignore the error due to random characteristics of vectors with small magnitudes. The mean corresponds to the average angle over the dots weighted with the number of drifters passing in a cell. In panels c and d, color represents the ratio of ADSC over surface drifter velocities. Size of dots represents the number of different drifters passing through a cell. Score written in the figure corresponds to the mean absolute error weighted with the number of drifters passing in the cells. The bottom row (e, f), depict the differences in magnitude and angle between the Geostrophy and Geo.+Ekman velocity vectors. Note the different color scales for these two panels than the top four.

The drifters, Geostrophy and Geostrophy+Ekman products have been gridded using the method prescribed above in section 2.4.2. The vector map (Fig 6) shows that drifters mainly sampled the shelf around Cape Farewell. The general pattern of the

current around the southern tip of Greenland shows a good consistency concerning the shape of the current between the ADSC and drifter-derived field. From a spatial pattern point of view, the along-shore current and the detachment of the coastal current from the coast at Cape Farewell are clearly visible and represented in all fields. To investigate more specifically this consistency, two quantities have been computed at each grid point: (1) the angle between the ADSC and the drifter-derived velocities and (2) the ratio of the ADSC over the drifter-derived velocities.

On the maps presenting angles between drifter velocities and ADSC directions (Fig 7 a and b), the angles are very small over most of the domain (smaller than $20°$) especially close to the shelfbreak where the direction of currents is a key point to resolve the shelf-basin exchanges. Some grid points very close to the coast present larger angles (larger than $60°$) but only few drifters passed inside these cells and the statistical significance is therefore smaller. The average angle over the region, weighted by the number of drifter traveling the cells, is: $14.5°$ for Geostrophy and $12.3°$ for Geostrophy+Ekman.

The maps showing velocity ratios (Fig. 7 c and d) depict areas of strong under and overestimation. The regions south of Cape Farewell and Cape Desolation are showing underestimation of about $50\%$, indicating that the ADSC fields have trouble resolving the zonal velocity in the coastal current. Again, this result is likely due to the large meridional separation between $1/4°$ grid cells that can be easily visualized in Fig. 7. Importantly, the magnitude of this error drops significantly at the shelfbreak, which is the key component for representing flux between the shelf and the ocean basin. So while the coastal current is likely not well-resolved south of Cape Farewell, the shelfbreak jet remains present. The mean absolute errors observed is $37.2\%$ for Geostrophy and $35.6\%$ for Geostrophy+Ekman. These percentages capture the magnitude of the errors, both in underestimation and in overestimation.

The difference between Geostrophy and Geostrophy+Ekman is small in these averaged fields. Indeed, the mean angle between the vectors is around $5.2°$ (Fig. 7 e) and the mean difference between velocities (Fig. 7 f) is about $7.1\%$ of Geostrophy+Ekman velocities. The main area where directions are different is on the West Greenland Shelf, around Cape Desolation with differences up to $12\text{-}15°$. Alternatively, we observe two hot spots where the magnitudes of velocities are different: South-East of Cape Farewell where Ekman contribution increases the magnitudes and North-West of Cape Farewell, around Julianehaab Bight, where the Ekman contribution decreases them.

The ADSC seem able to reproduce the current's direction with high accuracy, but the magnitude is underestimated. We suggest that this underestimation is due to the gridding of the along-track altimetry data that smoothes the velocities, and likely underestimates their true variance. The Ekman contribution seems to be a small contribution to the total velocity but improves both direction and magnitude consistently in both the along-shelf and across-shelf directions.

We compare $1/12°$ to $1/4°$ to evaluate the differences due to change in resolution (Fig 8). The $1/12°$ velocity grid shows more details and small-scale features (a). This product identifies the very strong velocity regions on the shelf south of Cape Farewell and on the shelfbreak south of Cape Desolation. Current direction and magnitude seem consistent and the mean kinetic energy (b, d) and eddy kinetic energy (c, e) fields show close values in general and in the energy hot spots as well. The $1/4°$ resolution therefore gets the right values at the right locations. In general, this exercise in testing various spatial resolutions of the gridded surface drifter velocity fields has shown us that future improvements to the gridded altimetry record from higher resolution products and the SWOT mission will likely improve the comparisons to the surface drifter velocities and allow for

more detailed studies of the circulation structure, but surprisingly, the 1/4° gridded fields capture the majority of the circulation

features important to the shelf-basin exchange in this region.

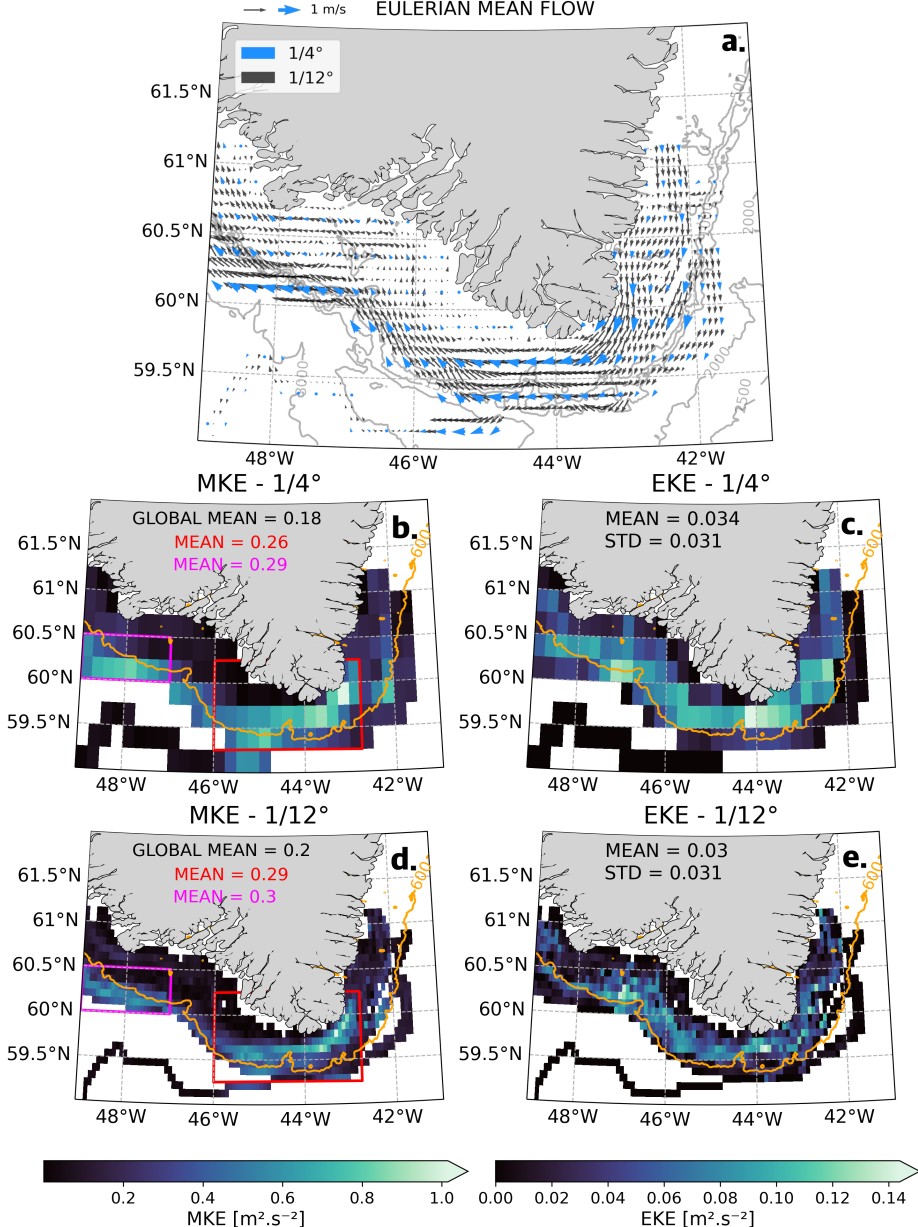

**Figure 8.** Comparison of gridded flows derived from drifter trajectories with varying resolution. (a) Comparison of 1/4° (blue) and 1/12° (black) vectors. Mean Kinetic Energy (MKE) maps obtained on 1/4° (b) and 1/12° (d) grids. Eddy Kinetic Energy (EKE) maps obtained on 1/4° (c) and 1/12° (e) grids. Values provided in panels b-e are the means over the color-coded boxes.

In this section, we presented a methodology to compute mean flows from ADSC that closely simulates the progressive sampling of the domain by the surface drifters, so that the Eulerian fields are averaged in the same spatial/temporal domain as the drifters. We also evaluated how sensitive this result is using a simple time average of the ADSC. Figure S3 (supplementary material) compares the Geostrophy and Geostrophy+Ekman velocity fields computed with these two different methods. They yield very similar fields, except in the region near the shelfbreak south of Cape Desolation. In this region, Geostrophy+Ekman computed with the 'drifter following' methodology clearly differs from the other fields. In our example, the methodology does not change much for Geostrophy only data set and the time averaged Geostrophy+Ekman field is very close to those Geostrophy fields. This could indicate that the Ekman contribution diminishes when averaged over a long enough period of time in this region where the synoptic storm time scale is around 3-5 days. The short time scale wind events likely play a role for the cross-shelf exchange as highlighted by the vector field obtained with the 'drifter following' method.

### 3.3   Observed and synthetic trajectories around Cape Farewell

ADSC velocities are now evaluated in the Lagrangian frame by applying the methodology described in section 2.3. The sensitivity of the skill scores to the duration of the particles' advection has been tested in repeating the experiment with various durations (from 2 to 14 days). We chose to use 3 days as proposed by Liu and Weisberg (2011) so that we could compare our values to theirs, as well as to avoid too short experiments which could be insignificant and too long where local information would be lost. Liu et al. (2014) applied this methodology to the evaluation of various remote-sensing product in the Gulf of Mexico and found a mean skill score of 0.50 in the open ocean and 0.41 on the shelf. The ADSC data they used were a combination AVISO (delayed time) + Rio2009 (MDT) + Ekman, which corresponds to the "Geostrophy + Ekman" product evaluated in our work. When the Ekman component is removed from their ADSC product, the open ocean skill score does not change but the shelf one decreases to 0.35. We can use their results as a reference to interpret the results obtained in the present work.

Here, the vast majority of our drifters traveled on the shelf, and we calculate a mean skill score of 0.47 for Geostrophy only and 0.50 for Geostrophy+Ekman (Fig. 9 a, b). Our skill scores on the shelf with the Ekman component (0.50) exceed the Liu et al. (2014) skill scores on the shelf (0.41) and are comparable to those obtained in the open ocean. Without Ekman, the mean score obtained with data mainly located on the shelf is 0.47. There is also a clear spatial signal in the skill scores, with an area of lower skill scores (red hatch) around Julianehaab Bight. These seem to correspond to the area of slowly eddying shelf flow described by Duyck and De Jong (2021). The average skill score associated is 0.38 for Geostrophy and 0.39 for Geostrophy+Ekman. This turbulent area, supposed to be more difficult to be reproduce because of its particular dynamics, shows almost similar results than Liu et al. (2014) with Ekman contribution on the shelf and even better results comparing to Geostrophy only. The rest of the shelf shows particularly high mean skill scores with 0.55 for Geostrophy only and 0.58 with the Ekman contribution. The results obtained here with the remote-sensing products are thus particularly good, especially considering the Ekman contribution. The impact of this contribution on the skill score is investigated by computing the difference between score obtained with both products (Fig. 9 c). Ekman contribution improves the main skill score by 0.03

and large improvements appear particularly on the East Greenland shelf for trajectories not located against the coast and south
of Cape Desolation close to the coast.

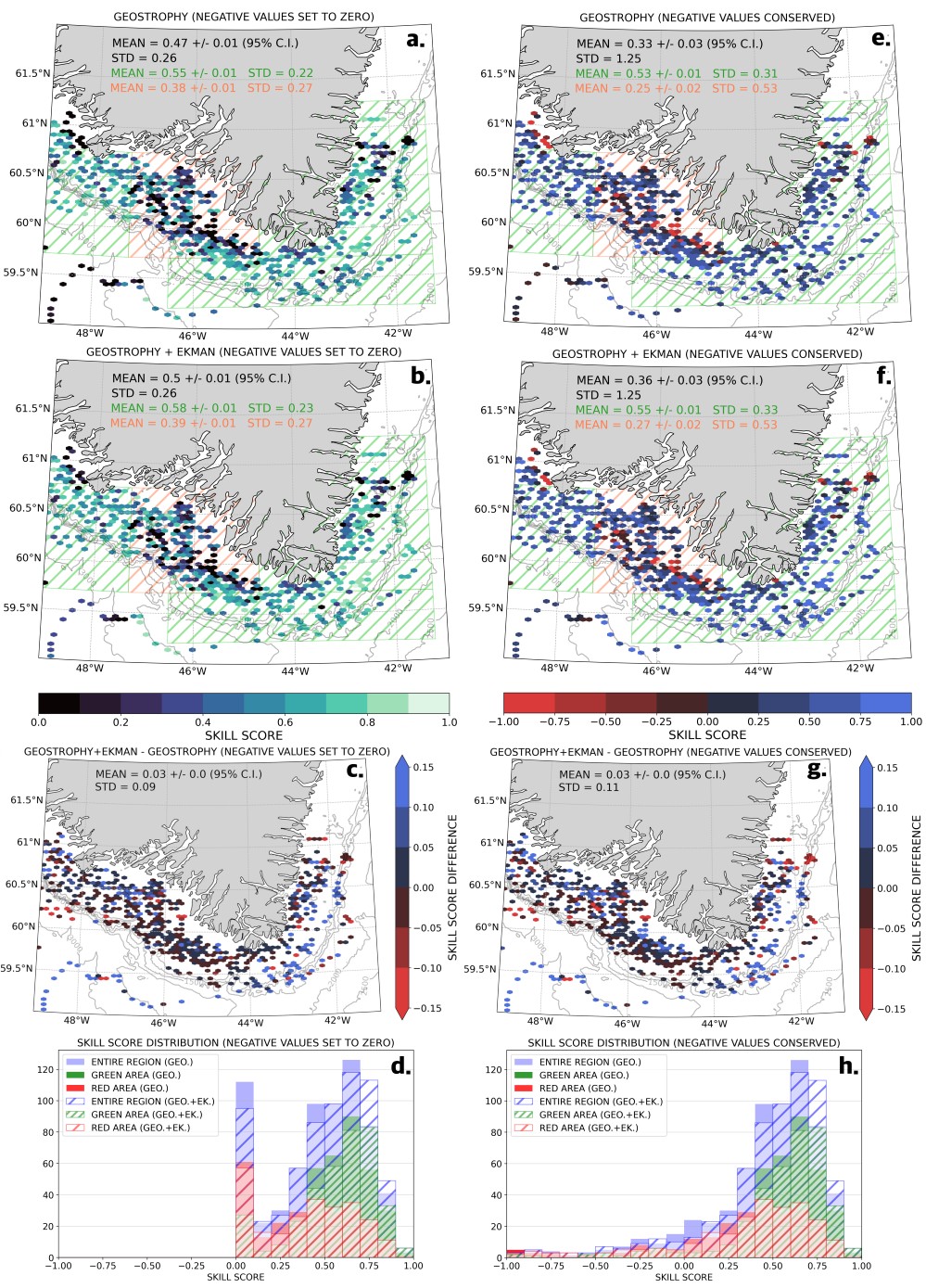

**Figure 9.** (Caption next page.)

**Figure 9.** (Previous page.) Synthetic trajectories evaluation for Geostrophy (a, e) and Geostrophy+Ekman (b, f). The dots represent the daily locations of drifters and the colors represent the score obtained for the trajectory computed with the 3 following days. An area with lower scores is hatched in red and the rest of the shelf with higher skill scores is hatched in green. Confidence intervals are computed using Bootstrap methodology. The left column (a, b, c, d) presents the results computed with the original Liu et al's methodology, which sets negative values to zero. The right column (e, f, g, h) is based on an alternative metric proposed in Révelard et al. (2021)), which conserves the negative values. The third row shows the skill score difference due to Ekman contribution, for the original (c) and the alternative (g) metrics. Blue dots denote increases in skill score when the Ekman velocities are included, and red dots indicate decreases. The histograms in the fourth row show the distribution of skill score for the different sub-regions (represented by colors) and using geostrophy (colored bars) or geostrophy+Ekman (hatched bars). The left figure (d) is for the original metric and the right one (h) presents the alternative score.

We extended the analysis by reproducing the results with the alternative metric (Fig. 9 d, e) suggested by Révelard et al. (2021), which conserved the negative values instead of replacing them by zero. The mean results decreased from 0.47 and 0.5 respectively for geostrophy and geostrophy+Ekman to 0.33 and 0.36, which remain good compared to skill scores obtained by Liu et al. on the shelf (0.35 and 0.41) without accounting negative values. The skill scores in the large shelf area, highlighted

by green hatches, remain very good and only decrease from 0.55 and 0.58 to 0.53 and 0.55 with this new metric. The results in the red shelf area present larger decreases from 0.38 and 0.39 respectively for geostrophy and geostrophy+Ekman to 0.25 and 0.27 but it remains on the order of Liu's results without negative values. Values using this metric reported in Revelard et al. (2021) from the Ibiza channel were considerably lower, with some regions hitting -0.6. The proportion of negative scores, proposed in Révelard et al. (2021) is also a very informative metric to understand the full picture. In our case for geostrophy

(geostrophy+Ekman), we obtained 13.1% (12.2%) of negative scores in the entire region, 7.3% (7.1%) in the red area and 3.4% (3.6%) in the green one. This low proportion of negative values explained the relatively small change of skill scores when accounting negative values, especially in the area of good consistency highlighted in green. The impact of Ekman contribution for this metric (Fig. 9 f) remain very close to the result obtained using the first metric (Fig. 9 c).

Finally, it is interesting to look at the distribution of scores for the different products, subregions and with the two metrics

(Fig. 9 g, h). Using the first metric (Fig. 9 g) we observed a peak between 0 and 0.1 that is not present with the second metric (Fig. 9 h) and simply represents the zeroing of all negative scores. In addition, we can observe that the score in the red subregion are more spread compared to score in the green subregion, especially considering the Revelard et al. metric. We can also observe that for the entire domain and the green subregion the number of scores smaller than 0.7 decreases and the amount of scores between 0.7 and 0.8, and in a lesser extent between 0.8 and 0.9, strongly increases. The low average score in the red

subregion is explained by the large variety of scores obtained with a combination of good and bad scores. Conversely, the good mean score in the green subregion is explained by the presence of almost only good scores.

From those results, we conclude that ADSC seem able to reproduce the trajectories of surface drifters in the region of Cape Farewell as well as they can reproduce trajectories in the open ocean in the Gulf of Mexico and Mediterranean. We expect the skill scores to be better for the open ocean than the shelf because the scales of motion are larger in the open ocean and the flow

is more geostrophic. Thus, it is impressive that the skill scores found on the shelf in the current study are comparable to the open ocean skill scores from Liu et al. (2014) and Révelard et al. (2021). Furthermore, our study is at higher latitudes where

the Rossby Radius of Deformation is smaller and thus the comparison of skills scores would favor more southerly latitudes. However, the tracks of polar orbiting altimeters converge at high latitudes so there is better along-track coverage, and we are also using an updated MDT data set in our analysis (2009 version of the CNES-CLS MDT compared to the 2018 version), both of which could contribute to this favorable comparison of our results.

## 4 Conclusions

The combination of all three steps demonstrates that the altimetry-derived surface currents are largely capable of recovering the spatial structure of the flow field on the South Greenland Shelf and can mimic the Lagrangian nature of the flow as observed from surface drifters. This good agreement is especially strong for the meridional velocities, likely due to the strong bathymetric constraints and the meridional orientation of the shelf in the area, as well as the higher spatial resolution in the zonal dimension of the gridded altimetry product. The Taylor skill scores drop for the along-shelf and across-shelf velocities, but they remain high considering that the shelf coordinate system accounts for the bathymetric steering of the flow. So, the skill scores for these velocity components are essentially estimating how well the flow field is characterized beyond the bathymetric control, and thus the skill scores are quite high given that assumption. ADSC show a particular ability to reproduce the direction of currents around Cape Farewell with errors on directions around 12° to 14° in average, depending on the product used.

Overall, the addition of the Ekman velocities to the Geostrophic product improved the comparisons, though not by as much as originally hypothesized and not for specific velocity components. It is likely that during high wind events this Ekman component is more noteworthy. Though our drifter observations were limited to a relatively short period in August and September of 2021, when winds are at their climatological minima, we note that the conditions experienced during this period in August and September of 2021 were noteworthy for their strong winds for this time of year. The drifters directly sampled upwelling and downwelling on the East Greenland Shelf, as well as upwelling on the West Greenland Shelf. Finally, the simulations of particle trajectories highlight the particularly good ability of ADSC to mimic the displacement of surface drifters in the region and by extension the displacement of water masses.

The main sources of error between the ADSC and the surface-derived velocities lie in the magnitude of velocities, especially the zonal velocities around the southernmost tip of Greenland. Here, the direction of the flow is well constrained, but the magnitude is about half of the observed velocity. This result is concerning, though the direction of the vectors remains high on the shelf, and the direction and magnitudes are well-resolved at the shelfbreak. As the shelfbreak circulation is critical for evaluating shelf-basin fluxes, we remain confident that the ADSC are doing a good job of tracking this exchange. Interestingly, the gridded products (step #2 in the methodology) seem to imply that the ADSC would be better suited to tracking the shelf-basin exchange than the exchange from the coastal current to the shelfbreak currents. But the Lagrangian simulations (step #3 in the methodology) imply that the ADSC are quite good on the SE Greenland Shelf, where we observe quite a bit of coastal current-to-shelfbreak exchange, especially under the initial upwelling-favorable winds. Thus, we conclude that the ADSC are capable of tracking the shelf-basin exchange, and one should specify exactly what is of interest in the coastal current-to-shelfbreak exchange prior to using the ADSC fields. Though our initial 'null hypothesis' was that the ADSC would

have trouble resolving the shelf circulation, the common result across all three steps of this proposed methodology was that the ADSC were capturing critical components of the circulation.

A large caveat of our results here has been that these results are specific to a roughly two-week period in Aug/Sep 2021 and may or may not be representative of the longer-term variability. To address this concern, we used a database of 34 drifters from the 6-hourly data set (Lumpkin and Centurioni, 2019) derived from Global Drifter Program (GDP) data that crossed onto the shelf from 1993-2021 (Supplementary Material Section 5 and figures S4, S5 and S6). The results from extending the temporal scope show a good coherence between surface drifters' trajectories and altimetry-derived surface currents from 1996 to 2020 in all seasons investigated, leading us to believe that this good correspondence is not specific to our brief study period, albeit with limited data to test. How far back in time one can reliably reconstruct the shelf circulation with ADSC, specifically as the number of altimeters decreases significantly prior to 2000, remains to be answered.

The results of this assessment pave the way for a long-term study of currents around the southern tip of Greenland based on satellite observations. They have the potential to improve our understanding of freshwater exchange between the shelf and the ocean basin by adding 30 years of observations to the results of modeling work, and possibly eliminating some model disagreements.

*Acknowledgements.* This work was funded by the National Science Foundation grant number 2047952. TERIFIC was funded from the European Research Council (ERC) under the European Union's Horizon 2020 research and innovation programme (grant agreement No 803140). The authors would like to thank Lucas Drumetz, Eleanor Frajka-Williams, Renske Gelderloos, Christophe Maes, Bob Pickart, Florian Sévellec and Pierre Tandeo for their input and guidance during this project. We would also like to thank the editor Erik van Sebille and the anonymous reviewers who helped us improve our manuscript considerably.

*Code availability.* The python codes to implement the presented methodology are available on the "ADSC-SVP-Comparison" repository (https://github.com/coquereau/ADSC-SVP-Comparison).

*Data availability.* The surface geostrophic current data set "Global Ocean Gridded L4 Sea Surface Heights And Derived Variables Nrt" is publicly available on the Copernicus Marine Service CMEMS website (https://doi.org/10.48670/moi-00149). The version including Ekman contribution named "Global Total Surface and 15m Current (COPERNICUS-GLOBCURRENT) from Altimetric Geostrophic Current and Modeled Ekman Current Processing" is also available on this website (https://doi.org/10.48670/moi-00049). Finally, CMEMS website also provides TOPAZ4 (Sakov et al., 2012) reanalysis in open access (https://doi.org/10.48670/moi-00001). "Global Drifter Program quality-controlled 6-hour interpolated data" from Lumpkin and Centurioni (2019) are publicly available (https://www.aoml.noaa.gov/phod/gdp/interpolated/data/all.php). The comparison with GDP data has been perfored on the reprocessed versions of altimetry-derived surface currents available on CMEMS as "Global Ocean Gridded L 4 Sea Surface Heights And Derived Variables Reprocessed 1993 Ongoing" (https://doi.org/10.48670/moi-00148) and "Global Total Surface and 15m Current (COPERNICUS-GLOBCURRENT) from Altimetric Geostrophic

Current and Modeled Ekman Current Reprocessing" (https://doi.org/10.48670/moi-00050). The deployed drifters from GFWE and TERIFIC are in the process of being incorporated in the GDP data set at the moment of submitting this manuscript.

*Author contributions.*  AC conducted the analysis and co-wrote the manuscript. NPF devised the study and co-wrote the manuscript.

*Competing interests.*  The authors have no competing interests to disclose.

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
