# Peer review of "Evaluating altimetry-derived surface currents on the South Greenland Shelf with surface drifters"

_EGUsphere, 2022_

## Author Comment (AC1)

**Authors response to Anonymous Referee #1**

**We would like to thank the referee for the time spent reviewing the manuscript in depth. From our point of view, the comments helped us to significantly improve our manuscript, in particular by refocusing the central objective of the article as suggested by the reviewer. We respond (in blue) to each of the reviewer's comments (in black) below and propose a revised manuscript, hoping to meet the expectations.**

general comments

1. The authors propose a more comprehensive approach for evaluating and intercomparing different Eulerian gridded surface current products against Lagrangian drifting data. It consists in evaluating gridded Eulerian products in 3 independent steps: a direct comparison of model velocities to drifter velocities, a comparison in the Eulerian frame of reference by transforming Lagrangian drifting velocities onto a Eulerian grid, and finally a comparison in the Lagrangian framework through the generation of model trajectories from eulerian velocities. These approaches are not novel as suggested in the title as each one is widely applied in the community but not necessarily altogether in one same study as done here. A novelty however is the complementary assessment of products using along and across shore velocities.

We agree with the reviewer that the individual steps in the framework we outline are not novel. We have instead rewritten much of the paper to focus on the validation of the altimetry fields around the southern tip of Greenland. This region is particularly important due to its potential for shelf-basin exchange and the supply of fresh polar water masses to the subpolar North Atlantic.

2. The authors illustrate the approach with a case study around Greenland comparing two altimetry-derived surface velocity products (geostrophy and geostophy+ekman), and conclude that both products reproduce well the shelf circulation of the region without entering too much in the detail as to why. The tone of the paper is ambitious, yet the manuscript does not contain major advances, and I feel that the manuscript would require significant modification for reaching the objectives it sets itself.

We agree that the dual motivations for our original submission were perhaps too ambitious for a single manuscript – a fact that was brought up by both reviewers. We have instead focused the text around the validation of altimetry-derived surface currents around southern Greenland. The result that the velocity field in this region is well-represented by the altimetry does advance the field because it allows the time-dependency of the flow field to be quantitatively assessed. To address why the altimetry fields perform well on the South Greenland Shelf, we have added text in the introduction about the structure of the shelf bathymetry that is amenable to reliable SSH retrievals, as well as added a new figure (Fig. S1, copied below) .This figure shows the along-track data coverage over the 10-day period in which the drifters were deployed, demonstrating the considerable coverage of the satellites that feed into the gridded product.

[Figure]

Caption: Along-track data coverage of six satellite altimeters incorporated into the gridded product during the 10 day period from August 15-24, 2021. Red dashed line in the left panel outlines the region shown in the right panel. Bathymetry is shown in the right panel.

specific comments

3. A concluding sentence of the introduction highlights the model errors in the region as a justification for such study, yet no model have been included in the investigation, solely altimetry derived products. The regional analysis system TOPAZ4 was included in step1 as a reference point, but not carried across in the following steps which is regrettable. One main purpose of carrying out such intercomparison work is to assess the skill of operational systems. Previous studies have evaluated altimetry products with operationnel systems (e.g. Dagestad and Rohrs, 2019). It would therefore be very valuable to assess TOPAZ4 in all 3 steps.

The focus of the paper is to evaluate the altimetry-derived surface currents in the region and thus we do not go into details about discrepancies between numerical models. There are many different reasons for why models may disagree with one another, notably sub-grid parameterizations, forcing fields, bathymetry, among others. Given the need for a central focus of this paper, we have decided to omit the reanalysis product TOPAZ4 from the majority of our analysis. The one exception is in Fig. 5, TOPAZ4 serves as a reference in our assessment of Taylor Skill Scores. This is done purely to compare the skill scores rather than an intent to assess the TOPAZ model.

4. The manuscript would benefit from a more exhaustive review of work carried out on the topic (and in the region), so to highlight the state of the art, present their limitation to justify the proposed approach.

We thank the reviewer for pointing out this omission and have now included a more robust introduction that includes information about the work in the region.

5. For step1, the methodology suggests to select the closest gridded product point to compare with the observation (daily average). I disagree with this approach and would recommend to interpolate the product value at the point of observation. It is not clear what resolution TOPAZ4 is, and for an approach to be robust a methodology should also be provided when comparing products of different resolution.

This is a valid concern that we had not considered, and thus we tested the sensitivity of our results to the interpolation methods and presented the results in section 3.1 lines 299-304:

"We tested the sensitivity of our results to the interpolation methods. When comparing the 'nearest neighbor' method with linear interpolation, the correlation improved in average by 0.02 the percent variance explained increased by 2%, but the normalized standard deviation and the Taylor Skill Score both decreased by 0.03 in average. Thus, interpolating reduced the variance in the gridded data while improving the correlations. The gridded altimetry is already a smoothed product from the along-track altimetry. We felt that retaining the natural variance is important. Given the similarity between the results, our conclusions from this first step are robust to this choice."

The resolution of TOPAZ4 is 1/4°, we thank the reviewer for pointing out this omission. We added this information in the section 2.2, line 105.

6. It is not clear how across and along shelf velocity are calculated, (this could be illustrated in figure1). The author mention smoothing the bathymetry, but without further detail (which bathymetry product, is the bathymetry common to all products?), how is the distance to the coast calculated? The authors introduce frequency spectrum analysis but do not carry it out in this study.

We have added text on lines 155-160 and equation 1 to further explain how we built the smooth shelfbreak and coastline and calculated the along and across shelf velocities. In addition, as suggested by the second reviewer as well in its comment 5, we zoomed out in the figure 1, which is now figure 2, and added the figure S2 in supplementary material (copied below) that covers the same area and helps to visualize the direction of along- and across-shelf velocities.

[Figure]

Comments for step2:

7. Lagrangian velocity is transformed onto Eulerian grid for comparison. The result is a map of average velocity for each grid cell where through drifter travelled for the period that the drifter covered. The spatial averaging is well explained and clear, what is not so clear is how the eulerian gridded product velocities have been selected to confront with each transformed lagrangian velocity. Let say if velocity values for a drifter travelling a given cell covers the temporal window not centered on the time window of the daily average gridded product, how is this addressed?

In this step, we build a time-mean surface flow from the drifters and compare it to the time-mean surface flow from altimetry. So we do not account for differences in when each drifter crossed through each grid cell.

8. A map of mean velocity vectors for each product, together with differences in magnitude and direction are proposed to evaluate the skill of products for the different subregions of the shelf. It would be interesting to measure second order statistics so to get a better feel of the product behaviour, and put uncertainties on the values (e.g. on skill score like Revelard et al. 2021).

We agree with this comment and thank the reviewer for this suggestion. We added in Fig. 9 (copied below) the standard deviation (STD) of scores over the different subregions (green and red hatched). In addition, we present in subplots g and h the distribution of scores for the different subregions, products and using the two metrics. These modifications help to better understand the different scores obtain for the two subregions. The region in green obtained only good scores and its STD is small while the region in red obtained more heterogeneous scores highlighted by a much larger STD. The interpretation of those distributions has been added on the manuscript on lines 393-400.

[Figure]

Comments for step3:

9. Statistics are done for the Liu skill score. A score that sets to 0 the negative values. Mean values of this skill score do not take into account the negative values, and may not be so robust when intercomparing models. Other studies have suggested alternative statistics for the skill score distribution such as the proportion of score > 0.6 (Revelard et al. 2021), which may be more informative.

We thank the reviewer for these suggestions and agree that the Liu skill score has some limitations, especially concerning the replacement of negative values with zero. However, we think that this skill score is still very informative because it allows us to directly compare our results with those obtained by Liu et al. . To investigate the sensitivity of our results to this skill score, we followed the reviewer's suggestion and reproduced the results using the metric derived in Revelard et al. (2021)

that conserves the negative values. In addition, we evaluated the proportion of negative values among our results. The figure 9 (copied above for comment 8) has been updated with the results from both metrics. Below is a copy of the paragraph interpreting these results, which has been added to the manuscript (lines 381-392):

The mean results decreased from 0.47 and 0.5 respectively for geostrophy and geostrophy+Ekman to 0.33 and 0.36, which remain good compared to skill scores obtained by Liu et al. on the shelf (0.35 and 0.41) without accounting negative values. The skill scores in the large shelf area, highlighted by green hatches, remain very good and only decrease from 0.55 and 0.58 to 0.53 and 0.55 with this new metric. The results in the red shelf area present larger decreases from 0.38 and 0.39 respectively for geostrophy and geostrophy+Ekman to 0.25 and 0.27 but it remains on the order of Liu's results without negative values. Values using this metric reported in Revelard et al. (2021) from the Ibiza channel were considerably lower, with some regions hitting -0.6. The proportion of negative scores, proposed in Révelard et al. (2021) is also a very informative metric to understand the full picture. In our case for geostrophy (geostrophy+Ekman), we obtained 13.1% (12.2%) of negative scores in the entire region, 7.3% (7.1%) in the red area and 3.4% (3.6%) in the green one. This low proportion of negative values explained the relatively small change of skill scores when accounting negative values, especially in the area of good consistency highlighted in green. The impact of Ekman contribution for this metric (Fig. 9 f) remain very close to the result obtained using the first metric (Fig. 9 c).

No information is given for the particle tracking software used.

We thank the reviewer for pointing this out. The particle tracking software used is Parcelsv2.0 (Delandmeter and Van Sebille, 2019, https://doi.org/10.5194/gmd-12-3571-2019). We added this information line 223.

Overall point for the case study:

10. This study investigates altimetry derived products for the Greenland shelf, with an assessment of Ekman component on improving the representation of shelf circulation. The study would benefit from a more in depth study of the processes. For example, evaluating the altimetry tracks, in particular their angle relative to meridional/zonal component, which could explain the better resolution of one component relative to the other. Relative to Ekman, questions are raised to explain why it may perform better than other regions. The Ekman component of the product could be assessed by investigating wind patterns of the regions, and see if a better parameterisation would improve results.

We thank the reviewer for these very interesting suggestions. Regarding the influence of altimetry tracks, we have added a figure of all the altimetry tracks used to calculate geostrophic velocities in the supplementary material (Fig. S1) and in response to comment 2. We agree that the angle of the tracks could have an impact on the ability to recover a particular velocity component. However, the figure shows that the tracks have different angles to the north-south directions, so it may not be straightforward to fully disentangle these relationships. Concerning the spatial differences in improvements due to Ekman contribution, there may be some very interesting connections to local wind regimes. However, investigating this question would require additional datasets and we believe that this is beyond the scope of the present ADSC evaluation study.

11. A paragraph is missing in the discussion wherein the 3 steps are brought together and their complementarity illustrated, as such it feels like 3 independent steps.

We agree with the reviewer and have added text in our conclusions section (lines 411-413) to explicitly say that it is really the combination of all three steps that helped us to build a comprehensive understanding of the consistency of ADSC around Cape Farewell.

12. The study does not address approaches raised in the community such as for example particle ensemble releases with diffusion terms to account uncertainties.

We explored the role of diffusion early on in our project, and two results led us to conclude that it was not worth considering. First, using only the surface drifter data, we calculated the Péclet number (added in Fig 1) and observed that the regions is largely dominated by advection. Secondly, we used the particle ensemble release methodology and observed large beaching of particles. We think this has to do with the combination of isotropic dispersion assumptions, a sharp land mask linked to the 1/4° resolution of the ADSC velocity fields, and the absence of mass conservation in ADSC Though we acknowledge there are methods to avoid beaching particles, they also introduce biases and uncertainties themselves and we were not convinced their benefits outweighed their costs. Given the results with the Péclet number, we decided to not use this method for this application.

13. I feel that the manuscript would be better pitched if the authors stated that they used a complementary approach to validate surface current (here geo and geo + ekman) illustrated with their Greenland case study, rather than presenting it as a new framework.

We agree with the reviewer and have significantly rewritten the text to address this concern.

---

## Author Comment (AC2)

**Authors response to Anonymous Referee #2**

**We would like to thank the referee for the time spent reviewing the manuscript in depth. From our point of view, the comments helped us to significantly improve our manuscript, in particular by refocusing the central objective of the article as suggested by the reviewer. We respond (in blue) to each of the reviewer's comments (in black) below and propose a revised manuscript, hoping to meet the expectations.**

General comments

1. The paper seems to have two goals. One is to propose a new method, another is to validate the satellite velocities on the southern Greenland shelf as the authors make it clear they have a special interest in this region. Rather than being a thorough methods paper, it seems like that authors needed to validate satellite data in this region and part of a process study paper was rewritten as methods paper. Right now these goals seems to be add odds, presenting an objective examination of the skill of a method, versus having a clear interest in the satellite data to come through the validation. This shows in several ways. The only gridded Eulerian data considered is satellite data. This is specified in the title, but needs to be clearer throughout the text. Some Lagrangian methods to deal with data are also not considered. The area where the data is compared is very limited, and is more or less confined to one type of flow regime (a fast boundary current). There are also issues with the latitude (rossby radius and oneven bin sizes) direction of shelf versus the grid. Nearly all aspects of time and variability are ignored, but are a main interest in studies of ocean currents. If this is to be a true methods paper then these issues needs to be addressed and the methods needs to be shown for a more widely representative area.

We agree that the dual motivations for our original submission were perhaps too ambitious for a single manuscript – a fact that was brought up by both reviewers. We have instead rewritten much of the paper to focus on the validation of the altimetry fields around the southern tip of Greenland. This region is particularly important due to its potential for shelf-basin exchange and the supply of fresh polar water masses to the subpolar North Atlantic. The result that the velocity field in this region is well-represented by the altimetry does advance the field because it allows the time-dependency of the flow field to be quantitatively assessed.

Line by line comments

2. Line 12. "A fluid parcel trajectory"

We agreed with the referee about this comment. We removed this paragraph after changing the focus of the paper.

3. Line 13. Please rephrase. It's not the physical systems that needs to be constrained, but the observations of.

We have deleted this paragraph in the updated manuscript.

4. Lines 40-45. What is completely missing from these questions is the time aspect. How well do methods 1 and 2 do in varying conditions/seasons? In 3, how well is variability captured? Since a large part of oceanographic research is not to understand the mean field, but to explain changes and trends, this is not something that should be neglected.

Data access is an obvious limitation for any project to validate the terrain in a region as remote as Greenland Shelf. We have the benefit of a relatively large data set of 34 drifter tracks for the Greenland shelf region. These drifters have covered the region in about two weeks. Therefore, we have a good amount of data to validate this specific period of summer 2021. To extend this assessment, we replicated the work with the GDP drifters by obtaining a comparable data set but moving over nearly three decades. From this dataset, we observed that the results appear consistent and not only valid for the summer of 2021, but they are clearly not sufficient to evaluate the evolution of accuracy throughout the seasons and years, which seems, with the data available in the whole scientific community, technically unattainable at this time. Indeed, if those data were available, it would not be worth considering the altimetry-derived currents.

5. Lines 63 to 69. It is not clear whether the described example applies to Fig 1. If it does, it would be helpful to zoom out in Fig 1 so one can more clearly see the shelf break and the chosen angles.

The figure 1 (now figure 2) does not seek to describe the velocity components but more illustrate the interpolation method along a drifter trajectory. However, we thank the reviewer for this suggestion, and we added the supplementary figure S2 (copied below) that covers the same area and show an example of the direction of along- and across-shelf velocities.

[Figure]

6. A question about this method. By limiting the comparison to discrete grid points, one might introduce artificial mismatched in the comparison. For example, if there is a clear front, or other dividing line between two flow regimes, the nearest grid point may be on the other side of the dividing line while the slightly further grid point may be in the same regime as the Lagrangian measurement. Allowing for interpolation between grid points would remote such artificial mismatches.

We agree with the reviewers that this is a valid concern that was also pointing out by reviewer 1 in their comment 5.  We tested the sensitivity of our results to the interpolation methods and presented the results in section 3.1 lines 299-304:

"We tested the sensitivity of our results to the interpolation methods. When comparing the 'nearest neighbor' method with linear interpolation, the correlation improved in average by 0.02 the percent

variance explained increased by 2%, but the normalized standard deviation and the Taylor Skill Score both decreased by 0.03 in average. Thus interpolating reduced the variance in the gridded data while improving the correlations. The gridded altimetry is already a smoothed product from the along-track altimetry. We felt that retaining the natural variance is important. Given the similarity between the results, our conclusions from this first step are robust to this choice."

7. Lines 70 to 74. While several of the comparisons are definitely useful, none of them are actually done in this paper.

In the original manuscript these comparison methods were suggested as general examples for use in this broad framework we outlined. Now that we have refocused the paper on the validation around Greenland, this part has been removed. However, we did use in our specific case three of the four methods proposed (standard deviation, root-mean-square error and the correlation coefficient). We did not used the frequency spectrum because it was not the objective of our comparison, but we added the evaluation of variance explained and the Taylor skill Score.

8. Lines 76 to 78. An important method that is left out here is clustering based on the Lagrangian data density (Koszalka and LaCasce, 2010). This is important because the data density itself provides important information on the flow field, not only in terms of fast and slow, but also in terms of convergence and divergence. This covers also the description of the problems in lines 81 through 83. However, the clustering method is more elegant than the methods proposed here, because it is not bound to fixed cell sizes.

We agree with the reviewer that this more elegant and better in lot of cases, especially when the particles or drifters are relatively homogeneously distributed on the domain. In our case, the sampling of the field by drifters is far from homogeneous, which is why we proposed a methodology to mimic the drifter progression in averaging the ADSC over time. In the figure below, we compared our methodology of resampling to the k-means clustering using 200 clusters containing in average 75 measurements each. While the field returned by the clustering is interesting to look at as it also shows in some sense the main pathways. It is not ideal in our case for a comparison to ADSC because of the drifters' sampling coverage. Indeed, we obtain a very high density of data point on the shelfbreak along the main pathways but almost no point in many locations. This is for instance the case of the east and south west of Cape Farewell while most of the 8 drifters launched on the shelfbreak (magenta in Fig 1d and 1e) crossed this area. We have integrated this reflection into the revised manuscript in section 2.4.2 lines 215-218.

[Figure]

9. Line 119-120. Rephrase the sentence "gridded Eulerian field" versus "time-averaged Eulerian field". I'm assuming the authors are still referring to Lagrangian data (density), but it's not clear from the sentence.

Lines 205-207. The way the two Eulerian fields were described here was confusing. The first Eulerian field mentioned is the one computed using the methodology we proposed, which is based on following the progressive sampling of the area by drifters. The second Eulerian field mentioned is obtained by simply averaging the ADSC over time during the entire experiment period. To clarify, we modify the sentence as follows: "To assess whether our results were sensitive to this procedure, we compared the results with this sub-sampled ADSC product with the mean of the entire ADSC fields over the same time period" We hope this revision clarifies our goals.

10. Line 137. Ect… Or, in the case of multiple trajectories, dispersion.

We agree with the reviewer this is an interesting metric to evaluate. To evaluate the importance of dispersion in our region we compute a Péclet number from the drifter trajectories and added the plot to the Figure 1. The Péclet number appears larger than one over the entire domain, with the limited exception immediately following the deployments, meaning that the advection largely dominates the flow. For this reason, we did not investigate the consistency regarding the dispersion.

11. Line 142. More attention could be given to describe what is good or acceptable skill score is here and elsewhere.

We acknowledge that it is difficult to determine what is a good skill score in the synthetic trajectory comparison step. To facilitate interpretation of these skill scores, we compared our scores to the scores obtained by the authors of the methodology (Liu et al.) in a large evaluation experiment in the Gulf of Mexico, as well as to Revelard et al. (2021).

12. Fig 3. Is the black dot at the start indicating a skill score of 0, or is it the start of the trajectory. In the latter case I suggest using a different symbol.

On the figure mentioned, which is now Figure 4 after rearrangement of the manuscript, the black circle indicates a low skill score due to a faster advection simulated by ADSC compared to the displacement observed with the drifter's trajectory. It is also the location where the drifter was deployed. To clarify this point, we added an orange star under the skill score dot to indicate the drifter deployment location.

13. Lines 172-189. While this is an interesting data set, it is very limited in time and space. Therefore it may not be the best to test these method. See also earlier comments regarding the time aspect that needs to be considered.

We agree with the reviewer's comment and have completely refocused the paper to address this concern.

14. Line 193 and 205. Would specify more clearly that the original satellite data does not have a daily resolution. The return period of a satellite over a certain track is 10 days. Therefore it is also not clear what the 6-hour resolution could add in information. What additional data would be included in this product that provides information on this time scale?

While we agree with the reviewer that the nominal repeat frequency of a single altimetry satellite is 10 days, there were six altimeters in various orbits, each with different repeat frequencies, during August 2021 when we deployed our surface drifters. The six-hour resolution of the "Geostrophy +

Ekman" product is primarily dictated by the temporal resolution of the wind field, which we clarify on lines 100-102.

15. Lines 244 and onwards. The reader would be helped with interpretation of these results if the authors first gave an explanation on what values represent poor, medium, good and excellent skill scores. Also, what significance level is used to determine if correlations are significant?

We agree with this comment. It is indeed difficult to determine what is a good Taylor Skill Score, this is why we used a reanalysis dataset (TOPAZ4) for comparison. To reinforce the conclusions, the skill score is completed by other metrics that are more easy to understand such as the correlation coefficient (where a typical coefficient between 0.4 and 0.6 presenting a moderate correlation, between 0.6 and 0.8 a high correlation and higher 0.8 a very high correlation), normalized standard deviation or root-mean-square error.

16. Lines 248 to 268. The difference in skill between zonal and meridional velocities is worrisome. If the explanation by the authors is correct, this suggests that the area used is not ideally suited to test this method. It would be more convincing to test the methods with regular boxes (either by design or at lower latitudes) and away from a shelf that is aligned with one direction of the grid.

We agree with the reviewer. This is also a reason that motivated us to refocus the paper on evaluating the altimetry-derived surface currents around the southern tip of Greenland.

17. Line 286. If there is good reason, than it's not surprising. Please rephrase this section.

We thank the reviewer to pointing that inconsistency out. We removed the work "surprisingly" in the revised manuscript, lines 284-285.

18. Fig 5. There quite a number of points with very low skill score and low correlation, which is worrisome and not commented on by the authors.

We have chosen to focus on the averages for the most part in the text because there were not clear systematic errors, and for every low value noted by the reviewer, the average was counterbalanced by a high value (by definition). To ensure that readers will not feel like we are glossing over poor comparisons, we have added text on lines 295-299 to acknowledge the distribution in scores.

19. Fig 6 and line 300. The difference in magnitude of some of these vectors is extremely larger. I am not surprised velocities from altimetry are lower, as a lot of averaging and filtering is applied to this data. I do not subscribe to the authors' interpretation that a factor 2 difference in magnitude is acceptable or very good.

We agree with the reviewer, and have rewritten this sentence in our revised submission. We are specifically commenting on the general structure of the flow field here, not the specifics which we outline in detail in Fig. 7 and the accompanying text.

20. Line 314 to 315. See the general comment on competing interests or focus in the paper. The large differences in the comparison should not be accepted because the authors have a special interest in the shelf break.

Once again, we agree that the example application around the southern tip of Greenland was not an ideal area to evaluate the metrics, especially when we mainly focused on the shelf-basin exchanges. The new focus of the revised manuscript, that does not seek to present and test a novel methodology but evaluate the consistency of ADSC around Greenland, in particular close to the shelf-break, should answer this comment.

21. Line 326. Variance is not addressed.

We agree that the variance has not been estimated in the second step of the methodology. However, we investigated the variance in the point-wise comparison (step 1) and all results indicated an underestimation of the variance of the velocities (Figure 5).

22. Line 370. Are skill scores of 0.55 really "particularly good"? Please keep in mind that these were also calculated for a very limited data set.

Compared to the skill scores obtained by Liu et al. (0.35 for Geostrophy and 0.41 for Geostrophy+Ekman), we think these scores (respectively 0.47 and 0.5) are very good. To improve our confidence in those results, we recomputed the scores following another metric (Revelard et al, 2021) as suggested by the first reviewer. We present those results on the Figure 9. The results obtained are again very good compared to the literature. Below is a copy of the paragraph interpreting these results, which has been added to the manuscript (lines 381-392):

"The mean results decreased from 0.47 and 0.5 respectively for geostrophy and geostrophy+Ekman to 0.33 and 0.36, which remain good compared to skill scores obtained by Liu et al. on the shelf (0.35 and 0.41) without accounting negative values. The skill scores in the large shelf area, highlighted by green hatches, remain very good and only decrease from 0.55 and 0.58 to 0.53 and 0.55 with this new metric. The results in the red shelf area present larger decreases from 0.38 and 0.39 respectively for geostrophy and geostrophy+Ekman to 0.25 and 0.27 but it remains on the order of Liu's results without negative values. Values using this metric reported in Revelard et al. (2021) from the Ibiza channel were considerably lower, with some regions hitting -0.6. The proportion of negative scores, proposed in Révelard et al. (2021) is also a very informative metric to understand the full picture. In our case for geostrophy (geostrophy+Ekman), we obtained 13.1% (12.2%) of negative scores in the entire region, 7.3% (7.1%) in the red area and 3.4% (3.6%) in the green one. This low proportion of negative values explained the relatively small change of skill scores when accounting negative values, especially in the area of good consistency highlighted in green. The impact of Ekman contribution for this metric (Fig. 9 f) remain very close to the result obtained using the first metric (Fig. 9 c)."

23. Line 386. "remarkably capable" is somewhat overstated given also lines 403- 406.

We agree about the overstatement and modify the sentence (lines 411-413) as:

"The combination of all three steps shown us that the altimetry-derived surface currents are capable of recovering the spatial structure of the flow field on the South Greenland Shelf and can mimic the Lagrangian nature of the flow as observed from surface drifters."

24. Line 407. "remain confident of tracking exchange" . There aspect of exchange is not addressed nor was it validated. Again, it shows a competing interest for the focus of the paper.

We agree that we did not evaluate this specific aspect of the circulation. However, we observed from the second step (Fig. 7) that the current is pretty well resolved, both in direction and magnitude, on the shelf-break. We also agree that there is competing interest for the focus of the paper, and we changed the focus to target the evaluation of ADSC around the south of Greenland.

25. Lines 417 to end. This is indeed a very large caveat and needs to be addressed. See also other comments.

(Line 441 to end) We agree that this is an important caveat, which is why we stated it. We also addressed it to the best extent possible by using the available data from the GDP program to

evaluate the consistency for different seasons and years . The results are detailed in section 5 of the supplementary material. To summarize, the results are as good as the results obtained with our own drifter dataset. From this, we cannot see any reason to conclude that the consistency is specific to the period of intense drifter sampling in summer 2021.

---

## Author Response (AR3)

**We would like to thank the referees for the time they devoted to this second revision of the manuscript. We respond (in blue) to each of the reviewers' comments (in black) below and propose a revised manuscript, hoping to meet the expectations.**

Arthur Coquereau and Nicholas P. Foukal

**Authors response to Anonymous Referee #1**

I am pleased with the modifications made to the manuscript,
The authors have addressed all the points raised during the first round of reviewing.
I would therefore recommend the manuscript for publication.

One minor point concerns the conclusion; a sentence should be added that relates to the questions raised in the introduction on the role of freshwater in the local to general circulation, and the suitability of satellite derived surface current product to help answering this question.

We are pleased that the revised manuscript has addressed the points raised in the first round of revisions. Regarding the point relating to the conclusion, we have added the following paragraph at the end of the conclusion (l. 450-453) to explain how these results will help to answer the questions raised in the introduction.

"The results of this assessment pave the way for a long-term study of currents around the southern tip of Greenland based on satellite observations. They have the potential to improve our understanding of freshwater exchange between the shelf and the ocean basin by adding 30 years of observations to the results of modeling work, and possibly eliminating some model disagreements."

**Authors response to Anonymous Referee #2**

The revised manuscript now has a clearer focus, which makes it easier to read and more informative. Many of the reviewer's comments were addresses. There are a few minor comments left. Some of the figure panels are very small and details are not visible (see comments below). Also, despite both reviewers requests, a qualification of the skill score is still missing. This should be easy enough to add in the text and it would add to the interpretation of the results to have this information.

Lines 135-144: There are three subsequent uses of 1), 2) and 3)> is not clear if the second two uses refers to the items described in the first enumeration or whether they are separate new enumerations.

The subsequent uses of numbers refer to the same three steps. For greater clarity, we have added the word "step" before the numbers (e.g. step #1).

Line 231: it stated that a skill score of 1 is nearly impossible to get, but it would help to have more information on what is considered a good or acceptable skill score. Further on the authors state that Lui et al found skill scores of such and such, but they do not say what qualification Lui et all gave these (bad, acceptable, good?). The general qualification could also be given around line 170, where the skillscore is introduced.

Liu et al does not define bad, acceptable, or good skill scores – they simply compare different models/products to their skill score by suggesting that models with higher skill scores better represent the flow. The skill scores are a useful tool to assess where and when the simulated particles are doing better and worse relative to one another, as well as relative to previous results. Here we use the same idea, in assessing where and when the simulated particles simulate the drifter trajectories well, as well as compare our results with those reported in Liu et al.

Line 273 "lower RMSE could be explained by the smaller magnitude of the across-shelf velocities which make the lower RMSE worse relatively to the magnitude of the velocity component" This sentence is hard to read and not very clear. Especially "make the lower RMSW worse" is vague, as low errors are generally good.

We agreed that our message was not clear in this formulation. The expression "signal/error ratio" explains more clearly what we meant here. We reworded the sentence as follows:

"this lower RMSE could be explained by the smaller magnitude of the across-shelf velocities, which imply a lower signal-to-error ratio" (l.273-274).

Fig 6. It is not possible to see the yellow arrows, or maybe they are hidden underneath other arrows. Similar comment for S3. Some colors do not show and the two blue colors that are used are not very distinct. If the differences are so small, maybe it is clearer to plot anomalies.

As the colors are not easy to distinguish, we have used another set of colors (avoiding yellow) that seems easier to see. Here, we have consciously chosen to present the magnitude rather than the anomalies, even though the superimposed arrow may be difficult to see. Indeed, in this case, it means that the vectors are almost identical, which is the main aim of this figure, to show the coherence between the vector fields.

[Figure]

Figures 8 and 9. The panels are very small and hard to see features and read the label text. Could these be made bigger?

We agree with this comment and have reorganized the sub-figures to increase their size.